# The lexicon of antimicrobial peptides: a complete set of arginine and tryptophan sequences

Sam Clark [1], Thomas A. Jowitt[2], Lynda K. Harris[1,3,4], Christopher G. Knight [5] & Curtis B. Dobson [1✉]

Our understanding of the activity of cationic antimicrobial peptides (AMPs) has focused on well-characterized natural sequences, or limited sets of synthetic peptides designed de novo. We have undertaken a comprehensive investigation of the underlying primary structural features that give rise to the development of activity in AMPs. We consider a complete set of all possible peptides, up to 7 residues long, composed of positively charged arginine (R) and / or hydrophobic tryptophan (W), two features most commonly associated with activity. We found the shortest active peptides were 4 or 5 residues in length, and the overall landscapes of activity against gram-positive and gram-negative bacteria and a yeast were positively correlated. For all three organisms we found a single activity peak corresponding to sequences with around 40% R; the presence of adjacent W duplets and triplets also conferred greater activity. The mechanistic basis of these activities comprises a combination of lipid binding, particularly to negatively charged membranes, and additionally peptide aggregation, a mode of action previously uninvestigated for such peptides. The maximum specific antimicrobial activity appeared to occur in peptides of around 10 residues, suggesting 'diminishing returns' for developing larger peptides, when activity is considered per residue of peptide.

[1] Division of Pharmacy & Optometry, School of Health Sciences, Stopford Building, The University of Manchester, Oxford Road, Manchester, UK. [2] Wellcome Trust Centre for Cell-Matrix Research, The University of Manchester, Oxford Road, Manchester, UK. [3] Maternal and Fetal Health Research Centre, The University of Manchester, St Mary's Hospital, Oxford Road, Manchester, UK. [4] St Mary's Hospital, Manchester Foundation Trust, Manchester Academic Health Science Centre, Oxford Road, Manchester, UK. [5] Department of Earth and Environmental Sciences, School of Natural Sciences, Michael Smith Building, The University of Manchester, Oxford Road, Manchester, UK. ✉email: curtis.dobson@manchester.ac.uk

Antimicrobial peptides (AMPs) are short amino acid sequences that kill or inhibit the growth of microorganisms[1], and which have diverse and broad mechanisms of action. They occur naturally as a component of innate immunity, and are produced widely by many diverse organisms[2,3], although they can also be designed de novo from natural or non-natural amino acids[4–6]. The presence of a high proportion of cationic and hydrophobic residues are two features commonly associated with AMP activity. Cationic residues (e.g. Arginine, R, Lysine, K or Histidine, H) are thought to mediate interactions with negatively charged bacterial lipids, whereas hydrophobic residues (e.g. Tryptophan, W, Phenylalanine, F or Leucine, L) mediate membrane association and/or damage. Naturally occurring AMPs are typically >15 amino acids in length, although active synthetic peptides <10 residues have been reported[5,7]. These have the advantage of low cost if used pharmaceutically, however shorter peptides typically have reduced antimicrobial efficacies relative to longer AMPs[7–9].

The hydrophobic and cationic residues most characteristic of shorter AMPs are W and R[10,11]. The aromatic and hydrophobic residue W in particular has been associated with antimicrobial activity[12–15]. Perturbing the bacterial cytoplasmic membrane is the most commonly studied mode of action for AMPs, including short cationic peptides[15]. This can occur through a number of proposed mechanisms, including the barrel stave and toroidal pore models, in which peptide monomers insert into the membrane and form structured pores, resulting in increased lysis, or the carpet model, in which peptides accumulate on the bacterial surface, causing stress on the membrane, which tears causing lipid removal[5,16].

Peptide aggregation prior to contact with membrane may also be required for activity for some AMPs, although this is at present uncertain[17]. We have previously reported that the self-aggregating amyloid β (Aβ) peptide associated with senile plaques in Alzheimer's disease (AD) is upregulated in response to infection[18], and others report that it may function as a conventional AMP[19,20]. More broadly, although some degree of peptide aggregation is likely to occur for most hydrophobic AMPs in aqueous solution, surprisingly the role of aggregation in conferring or inhibiting antimicrobial efficacy remains largely unstudied.

We have previously examined the impact of systematic and complete substitution of all eight hydrophobic L residues in apoE-derived cationic AMPs with any other single natural amino acid. Of these, W substitution clearly yielded the most efficacious and broadly active peptide[12], and reflected increased membrane perturbation by the W variant[13]. In other AMPs, W and R residues have been reported to associate with each other through delocalised π-electrons, with the side-chains of both residues lining up, allowing hydrogen bonding between the R residues and the solvent. This allows R (but not the other strongly cationic natural amino acid K) to insert into a lipid membrane by masking its charge within the benzene ring of W[14].

W and R are the two natural residues most closely associated with AMP activity perhaps due to their being the most hydrophobic or cationic residues, respectively[21]. Although indirect evidence supporting the importance of W and R in conferring antimicrobial activity has emerged, to date their role has not been addressed directly. More specifically, the antimicrobial activity of the native W or R amino acids and the development of efficacy as these residues are systematically combined in peptides of increasing size and in most or all possible permutations has not previously been examined. Additionally, it is unclear what mode of action would predominate in this emerging family of peptides. Peptide 'combinatorial space' is very large (considering only 10-residue peptides and the 20 natural amino acids there are $20^{10} = 10^{13}$ possible sequences; i.e. around hundred times the reported

number of neurons in the brain). No studies have previously investigated activity within a complete set of peptides, comprising all possible peptides generated according to a specific formula, from individual residues upwards. However, a focus on W and R as key determinants of antimicrobial activity, drastically reduces the space of possible sequences (for two amino acids there are only 254 possible peptide sequences across all lengths from one–seven residues), enabling such a comprehensive approach.

In the present study we have synthesised a peptide library comprising all possible sequences up to seven residues in length that can be produced from W and R. We have characterised the emergence of antimicrobial efficacy against medically relevant organisms, assessed haemolytic activity (as an indicator of mammalian cell toxicity) within this peptide space and identified size and sequence characteristics associated with activity. We examined the mode of action of these peptides and established whether this correlates with peptide aggregation and/or affinity for negatively charged or neutral membranes. These observations enable us to make several broad conclusions about AMPs constructed from hydrophobic and cationic residues alone.

## Results

**Antimicrobial activity**. We considered all possible sequences of W and R, up to seven residues long. These two amino acids are closely associated with AMP activity, however, the shortest 'peptides' assayed were the individual amino acids, which we expected to be inactive. In this way we uncovered the development of antimicrobial activity in fine detail. We obtained values for microbial inhibition (half maximal inhibition, $IC_{50}$; Fig. 1a, b) and microbicidal activity (minimum biocidal concentration, MBC; Supplementary Fig. 1a) for 252 of the 254 sequences in this space (the remaining two sequences were so hydrophobic that they could not be synthesised—we therefore considered these to be in effect inactive). Organisms tested were the gram-positive bacterium *Staphylococcus aureus*, the gram-negative bacterium *Pseudomonas aeruginosa* and the yeast *Candida albicans*.

Antimicrobial activity increased with peptide length for all organisms. The minimum length necessary for measurable inhibitory activity was four residues for *S. aureus*, or five for the other two organisms (Fig. 1a). Similarly, MBC values could only be determined for peptides with a minimum of four (*S. aureus*), six (*P. aeruginosa*) or five (*C. albicans*) residues (Supplementary Fig. 1b). Not only was antimicrobial activity only seen in longer peptides, the extent of activity was positively correlated with length (Fig. 1b and Supplementary Fig. 1d).

Overall, antimicrobial activity was highest for *S. aureus* > *C. albicans* > *P. aeruginosa*, with most peptides in the heptamer group (which comprised approximately half the library) showing activity. Minimum heptamer $IC_{50}$ concentrations were 1.9, 10.3 and 15 μM for *S. aureus*, *C. albicans* and *P. aeruginosa*, with 92.2, 81.3 and 60.2% of heptamers showing measurable $IC_{50}$ concentrations against these three organisms (Fig. 1a). MBC values followed a similar pattern with the minimum MBC values for heptamers being 9.6, 18.7 and 42 μM for *S. aureus*, *C. albicans* and *P. aeruginosa*, respectively, with 60.2, 60.2 and 10.2% of heptamers showing measurable MBCs against these three organisms (Supplementary Fig. 1a). We found a strong correlation between $IC_{50}$ and MBC for each of the three organisms for the entire peptide set (*S. aureus*: $\rho = 0.73$, $P < 2.2 \times 10^{-16}$, $S_{129} = 101042$; *C. albicans*: $\rho = 0.93$, $P < 2.2 \times 10^{-16}$, $S_{114} = 17027$; *P. aeruginosa*: $\rho = 0.78$, $P = 2.375 \times 10^{-5}$, $S_{19} = 384.42$).

The absolute activities of each peptide molecule expressed by plotting the inhibitory concentration in μM presented in Fig. 1a, b is helpful in understanding how activity increases with peptide length. A further insight is provided by considering 'specific

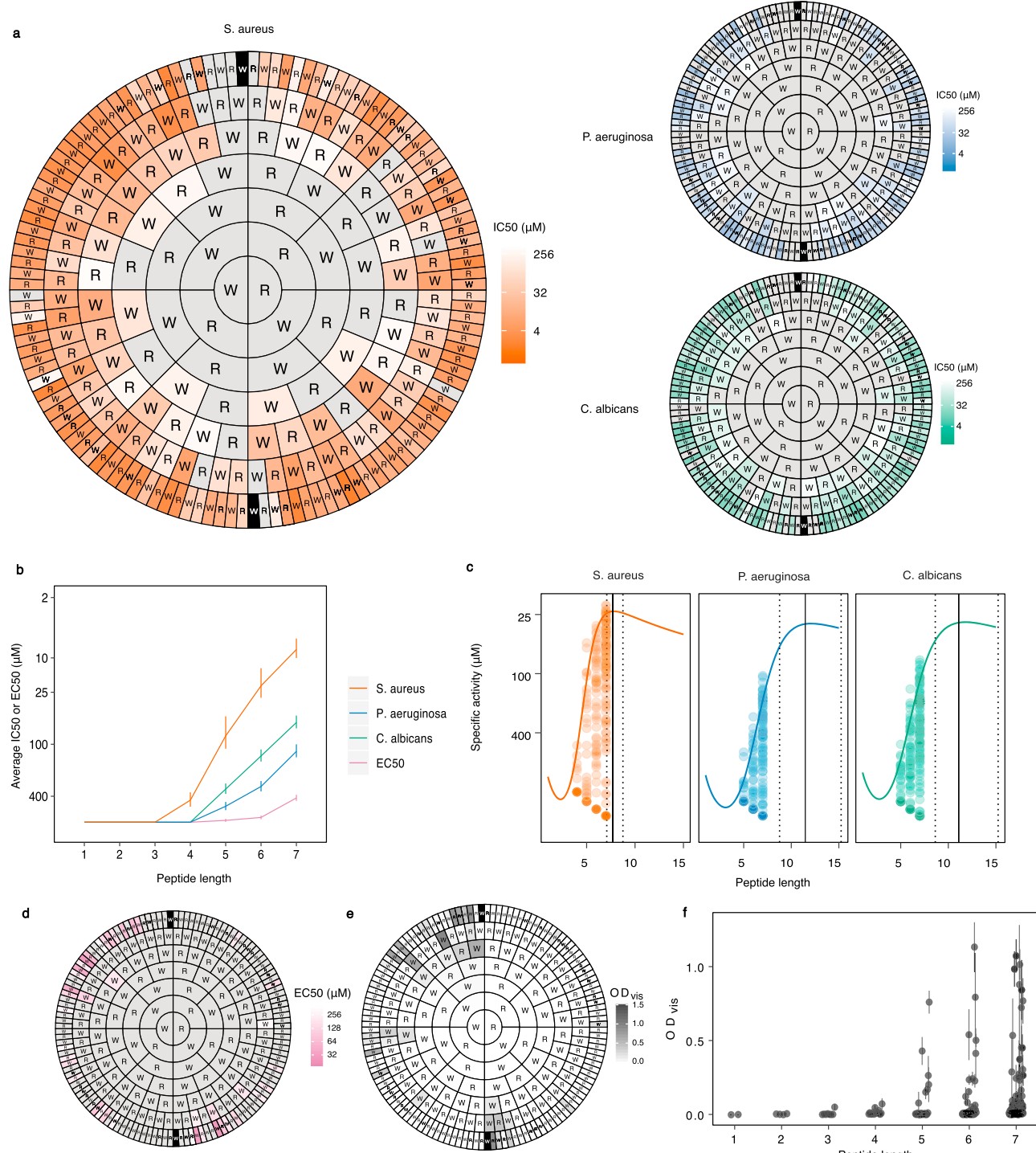

**Fig. 1 Antimicrobial activity for the complete set of peptides comprised of W and R up to 7 residues long. a** Activities are represented for peptide sequences arranged using a concentric ring chart system (referred to as 'Harris-Clark diagrams'). Each peptide can be identified by reading the sequence from the N-terminal residue in each ring towards the C-terminal residue at the centre of the chart; inhibitory activity ($IC_{50}$) is indicated for that peptide by the colour of shading in the outermost compartment (i.e. the compartment identifying the N-terminal of each peptide). The three separate Harris-Clark diagrams show the inhibitory activities against *S. aureus*, *P. aeruginosa* and *C. albicans*. Grey sections represent peptides which did not exhibit an $IC_{50}$ within the range of concentrations assayed (0.8–400 μM). The two peptides which could not be synthesised are indicated by black sections with white lettering. Similar plots for MBC are shown in Supplementary Fig. 1. **b** Effect of peptide length on harmonic means and standard deviations for $IC_{50}$ and $EC_{50}$. Similar plots for MBC are shown in Supplementary Fig. 1. All error bars shown are ±s.d. ($n = 2, 4, 8, 16, 32, 64$ and 126 peptides for lengths 1–7, respectively) **c** Specific activities of each peptide plotted against length. Quantile regressions (90% quantile), represented by coloured lines, show the trend of specific activity with length. The solid vertical lines indicate the predicted peptide length with the greatest specific activity, and the dotted lines indicate the 95% credibility interval for this prediction generated by sampling from the posterior distribution of a Bayesian model (see Methods). The distribution of haemolytic activity for individual peptides is shown in (**d**) and of turbidity of 800 μM stock solutions ($OD_{vis}$) in (**e**), whereas (**f**) shows $OD_{vis}$ of the same stock solutions plotted against peptide length.

activity', i.e. how the peptide size influences the increase in activity from adding on further residues. In this way we could determine whether this assessment of specific activity (activity per residue) has a peak, if so suggesting that then adding further residues will lead to lower overall activity per unit mass of peptide. We estimated this length using data for the complete peptide library as shown in Fig. 1c. The peptide lengths with maximal specific activity were estimated at 8 (7.08–8.75, 95% CI) residues for *S. aureus*, and 11 residues for *P. aeruginosa* (8.77–15.2) and *C. albicans* (8.63–15.2).

**Haemolytic activity**. Haemolytic activity was detectable for fewer peptides than antimicrobial activity; only 40 peptides had an $EC_{50}$ within the concentration range assayed. Only peptides longer than four residues in length displayed measurable haemolytic activity and, as observed for antimicrobial activity, there was a positive association between haemolytic activity and peptide length (Fig. 1b, d and Supplementary Fig. 1c). These values can be used to normalise the antimicrobial activity ($IC_{50}$) to give a 'therapeutic index' indicating the increase in efficacy of the peptides against a particular microbe relative to mammalian cells (Supplementary Fig. 1e).

**Solution turbidity**. Measurement of the turbidity ($OD_{vis}$) of each peptide solution, which was used as a means of broadly assessing the extent of peptide aggregation, revealed that some peptides with five or more residues, and in particular those containing higher numbers of W residues, exhibited increased turbidity (Fig. 1e, f). This may reflect reduced solubility or an increased propensity to aggregate, thereby producing large visible structures. These peptides typically showed low activity, with the exception of the cluster of peptides ending with 'WWRWW'(in the upper left of Fig. 1e). These sequences exhibited potent activity, particularly against *S. aureus*, despite producing turbid solutions.

**Sequence-activity relationships**. The landscapes of sequence-activity relationships (Fig. 1a and Supplementary Movie 1) contain a 'corridor' of inactivity (from the centre to the uppermost region of the circles in Fig. 1a) corresponding to peptides comprising only W or only R. This illustrates the importance of a mixture of the two residues underpinning activity. Efficacy was typically highest against all organisms for heptamer peptides containing 3R and 4W residues, although this ratio may be less critical for *S. aureus*, as the peak was broader (Fig. 2a and Supplementary Movie 1). Nonetheless, small differences in sequence could be associated with large differences in activity, for instance, the heptapeptide WWWRRRW is highly active ($IC_{50} = 2.9 \pm 0.3$ μM) but changing one residue, to WWWRRRR increases the $IC_{50}$ 65-fold (Fig. 1a and Supplementary Movie 1). Such variability in antimicrobial activity with sequence can be characterised by calculating the roughness to slope (r/s) ratio of the peptide landscape at each length (Fig. 2f). This value increased with length, indicating that the spread of possible activities becomes greater with increased peptide length and complexity.

Because of this landscape complexity, we applied a machine learning feature selection algorithm to identify sequence features associated with antimicrobial activity in the complete peptide library (Supplementary Table 1). As expected from Fig. 2a, a key feature important for activity against all of the organisms studied was W content. Typically, for penta-, hexa- and heptapeptides, those with ~60% W content exhibited the lowest $IC_{50}$ concentration against all organisms (Fig. 2b), with a similar trend apparent for the MBC (Supplementary Fig. 2a). Heptapeptides containing 70–80% W had the most potent haemolytic activity (lowest $EC_{50}$ values) (Fig. 2e). The $OD_{vis}$ typically increased with W content,

suggesting increasing hydrophobicity was associated with either insolubility or propensity to aggregate (Fig. 2d).

Another key feature associated with antimicrobial activity was presence of multiple adjacent W residues (Supplementary Table 1). While W singleton (isolated W residues in the sequences) showed no correlation with antimicrobial activity, there were positive correlations of antimicrobial activity with both duplet (WW) and triplet (WWW) content (Fig. 2c). These findings suggest clustering of W residues as doublets or triplets is important for antimicrobial activity. No similar relationships were observed for R residues (Supplementary Fig. 2b).

We considered the possibility that endogenous cleavage of the peptides by proteases might affect activity. As almost all naturally occurring proteases would cleave our peptides either just after W or R, indicating extent of cleavage by different proteases would simply depend on the amount of R and W in each sequence. One exception was pepsin, which cleaves after W depending on the position of R within the sequence. This was therefore examined using in silico analysis, in which the number of pepsin fragments likely from each of our peptides was plotted against antimicrobial activity (Supplementary Fig. 3a). We identified correlations between likely number of pepsin fragments and activity, which varied among organisms and peptide lengths. We also considered whether hydrophobic moment might be associated with activity, and found evidence for weak positive correlations (Supplementary Fig. 3b).

**Heptapeptide aggregation**. $OD_{vis}$ measurements (Fig. 1f) suggest that more aggregation occurs in some longer peptides. A model of these indicates that, having accounted for the effect of length, the residual correlation between peptide $OD_{vis}$ and $IC_{50}$ against all organisms, is weakly negative (−0.09 [−0.17 to −0.01 95% CI]). In other words, this aggregation is, if anything, beneficial to peptide activity. We therefore used dynamic light scattering (DLS) to measure the typical size of aggregates among the longest (7-residue) peptides. Heptapeptides fell into three classes: those producing small aggregates (diameter < 10 nm), moderate aggregates (10 nm < diameter < $10^4$ nm) or large aggregates (diameter > $10^4$ nm) (Fig. 3a). The $OD_{vis}$ distributions for heptapeptides differed noticeably among aggregate classes, with those forming large aggregates having significantly higher $OD_{vis}$ values than heptapeptides producing small aggregates (Fig. 3b). This is likely to be due to larger aggregates scattering more light, resulting in higher measured turbidity. Percentage W content was also positively associated with aggregate size (Fig. 3c). Tendency to produce large aggregates was negatively associated with antimicrobial activity, compared to peptides in the other two aggregation states, irrespective of microbial species (Fig. 3d). A similar association was found with MBC (Supplementary Fig. 4). Haemolytic activity showed no significant relationship with aggregation state (Fig. 3e).

To test whether heptapeptide aggregation was concentration-dependent and reversible, aggregation was re-examined by analytical ultracentrifugation (AUC), using 80 μM dilutions of the three heptapeptides that exhibited substantial aggregation at 800 μM (as measured using DLS). Each peptide formed aggregate structures <5 kDa, suggesting that, at most, tetrameric aggregates (~4.8 kDa) are present (Fig. 3f). This concentration-dependent effect on aggregate size was corroborated by transmission electron microscopy of WRWWWWW. Substantial visible aggregates were present at 800 μM but not at 80 μM (Fig. 3g). Together these findings suggest that heptapeptide aggregates disperse at lower concentrations.

**Membrane binding of heptapeptides**. Negatively charged bacterial membranes are frequently considered to be a specific

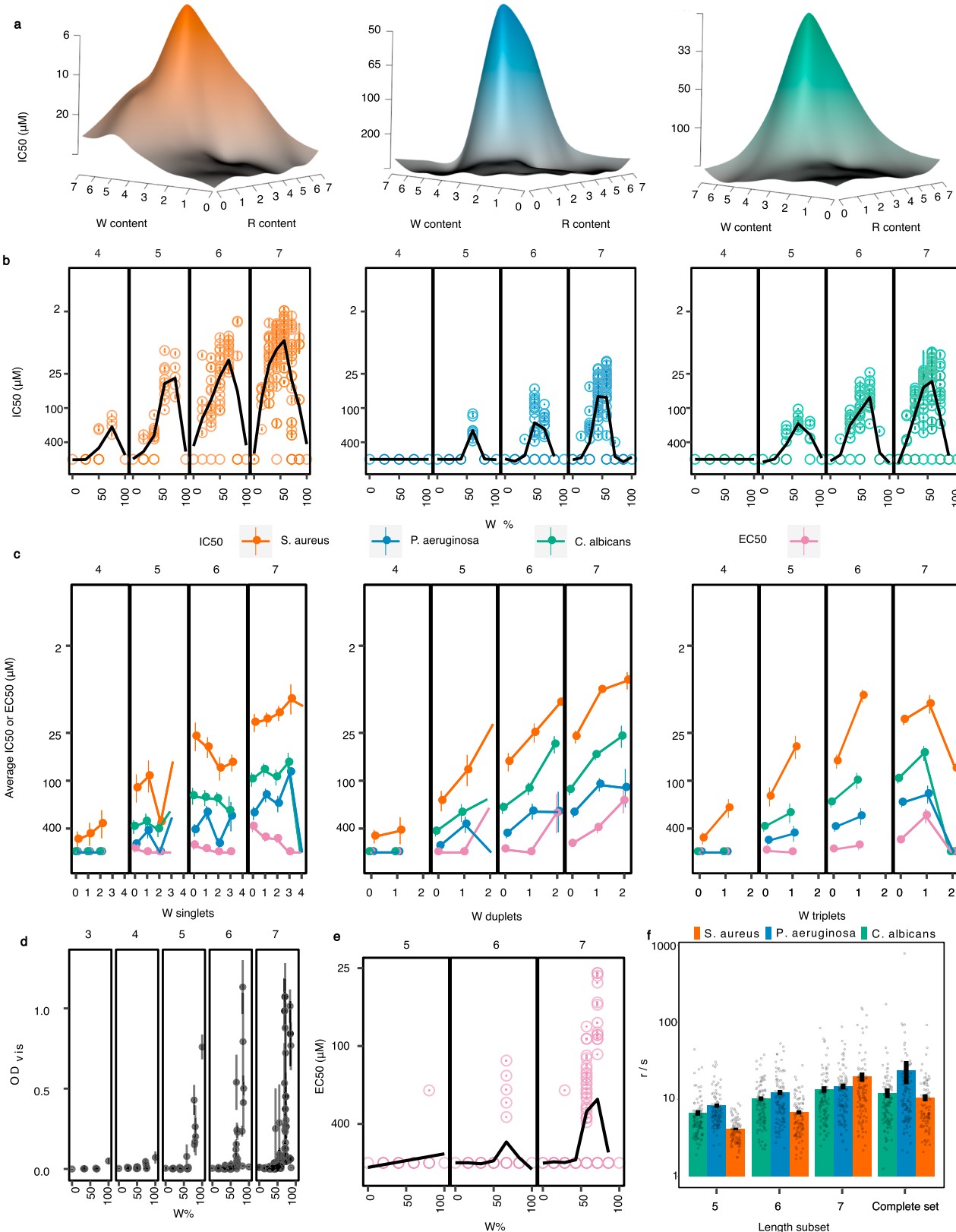

molecular target for positively charged AMPs. We therefore used a quartz crystal microbalance with energy dissipation (QCMD) to test the membrane association of 29 heptapeptides (comprising 20 pairs of heptapeptides possessing the largest differences in activity whilst differing by only a single substitution in the sequence, Supplementary Fig. 5) with model negatively charged (anionic) or neutral lipid membranes. These compositions represent typical bacterial or mammalian cell membranes, respectively[22] The mass of bound heptapeptide was highest for neutral membranes for those heptapeptides with either four or five W residues, and for the anionic model membranes for heptapeptides with 4 W residues (Fig. 4a). A significant positive

**Fig. 2 Effect of various peptide primary structural features on activity. a** Activity (IC$_{50}$) presented as a three-dimensional representation illustrating (as a landscape) the influence of the number of W and R residues on activity against the three different organisms. **b** Inhibitory activity plotted against percentage W residues within the sequence, faceted by peptide length (indicated at the top of each sub panel). The black spline through the data indicates the average activity for each peptide length. Error bars are within the markers and are ±s.e.m. ($n = 4$). Similar plots for MBC are shown in Supplementary Fig. 2a. **c** Analysis of average inhibitory and haemolytic activities for peptides with different numbers of isolated W singlets (W), duplets (WW) and triplets (WWW), faceted by peptide length (indicated at the top of each sub panel). Error bars shown are ±s.e.m. Similar plots for R, RR and RRR are shown in Supplementary Fig. 2b **d** Effect of percentage W content on average OD$_{vis}$, faceted by peptide length (indicated at the top of each sub panel). Error bars shown are ±s.e.m. ($n = 4$). **e** Haemolytic activity plotted against percentage W residues within the sequence, faceted by peptide length (indicated at the top of each sub panel). The black spline through the data indicates the average activity for each peptide length. Error bars are within the markers and are ±s.e.m ($n = 4$). **f** Roughness analysis, indicating the extent of variability of activity across the three different antimicrobial activity landscapes. Error bars shown are ±s.e.m. ($n = 100$ in all cases).

correlation was observed between the distribution of R through the sequence (positional standard deviation) and heptapeptide mass bound to anionic model membranes ($\rho = 0.49$, $P = 0.004$, $S_{27} = 3040.6$) (Fig. 4b). Interestingly, the same correlation was not seen between the distribution of R and the mass bound to neutral membrane ($\rho = 0.04$, $P = 0.83$, $S_{27} = 5241.7$).

A positive relationship between activity and binding to both neutral and anionic model membranes was also identified (Fig. 4c–e), although the correlation was stronger for anionic membrane binding (*S. aureus*: $\rho = 0.61$, $P = 2.5 \times 10^{-4}$, $S_{27} = 9609.9$; *P. aeruginosa*: $\rho = 0.74$, $P = 2.1 \times 10^{-4}$, $S_{27} = 10437$; *C. albicans*: $\rho = 0.59$, $P = 4.4 \times 10^{-4}$, $S_{27} = 9488.6$) than neutral membrane binding (*S. aureus*: $\rho = 0.39$, $P = 0.26$, $S_{27} = 7608.2$; *P. aeruginosa*: $\rho = 0.26$, $P = 0.14$, $S_{27} = 6897.5$; *C. albicans*: $\rho = 0.50$, $P = 3.6 \times 10^{-3}$, $S_{27} = 8207.6$). In contrast, the correlation of haemolytic activity with neutral membrane binding ($\rho = 0.76$, $P = 1.45 \times 10^{-6}$, $S_{27} = 9636.7$) was stronger than that with anionic membranes ($\rho = 0.32$; $P = 0.06$, $S_{27} = 7952.4$) (Fig. 4f). Bacterial membranes are more negatively charged than mammalian membranes, and thus our data are consistent with antimicrobial activity for these peptides relating to their binding negatively charged membranes and their mammalian cell cytotoxicity originating in their neutral membrane binding.

**Morphological changes induced by heptapeptides.** Finally, we compared the effect of an inactive heptapeptide (RRRRRRR) and one of the most active heptapeptides (RWWRWWR) on bacterial morphology after a short exposure. Scanning electron microscopy of *P. aeruginosa* revealed no discernible change in morphology between the control bacteria and those treated with the inactive RRRRRRR, at 80 or 400 µM for 1 h (Fig. 5a). However, exposure to the highly active heptapeptide RWWRWWR (80 µM; 1 h) resulted in the formation of spheroid objects on the surface of the bacteria, indicative of membrane blebbing (indicated by the black arrows in Fig. 5a. Furthermore, treatment of bacteria with a 400 µM solution of RWWRWWR for 1 h resulted in a dramatic reduction in the number of intact bacteria visible in the sample, and the presence of large amounts of smaller irregular material (indicated by the white arrows in Fig. 5a).

Atomic force microscopy of *P. aeruginosa* treated with 100 µM RWWRWWR (for 2–45 min) highlighted significant morphological differences through treatment. Mean bacterial width decreased by a third and height doubled after only a 2 min exposure (Fig. 5b, c). These SEM and AFM findings together suggest that peptide activity involves rapid perturbation of bacterial membranes, firstly changing cell shape, then later inducing observable morphological irregularities, which, at higher concentrations, results in bacterial cell lysis.

## Discussion

Our approach of characterising the biological activity of all possible peptides up to seven residues comprising R and W, enables us to make some general conclusions about the rules governing the activity of synthetic AMPs comprising cationic and hydrophobic residues. Although previous studies have provided anecdotal examples of the impact of sequence changes on the activity of several such peptides[23], we examine these effects using a complete set of all peptides possible within a theoretical universe, and therefore we can more reliably make broad conclusions at least for the peptide class studied. It should be noted that our aim was to provide fundamental insights into the rules governing the development of activity in this class of peptides, and we did not expect to identify highly potent sequences suitable as potential clinical lead compounds[24].

We established the minimal length of peptide required for activity to be 4–5 residues (to obtain a measurable IC$_{50}$ value) and 4–6 residues (to obtain an MBC) for the three organisms tested. Although a measurable MBC is a key attribute of an AMP, as we focused on investigating the development of antimicrobial activity in this peptide set, we also considered IC$_{50}$ values to be important as an indicator. Indeed we found that IC$_{50}$ concentrations were likely to be a good indicator of activity overall, since the IC$_{50}$ and MBC against each organism was highly correlated for peptides in the set which exhibited activity in both metrics.

In general, activities were broadly similar for all three organisms, although more peptides showed activity against *S. aureus* and *C. albicans* than against *P. aeruginosa* (in particular when assessed by measuring MBC), perhaps reflecting greater resistance of the gram-negative organism conferred by its outer membrane. These findings might be confirmed by extending testing to a broader range of organisms. The minimal peptide length we uncovered is consistent with other reports in the literature for isolated examples of peptides, where activity typically only occurred for sequences of naturally occurring amino acids >4–5 residues in length[2]. One previous study has reported detectable activity for dimer peptides[25]; however such activities would not of been measurable in our study due to the extremely high concentration of peptide required (in the mM range, where we tested up to 400 µM). A minimum length of two residues has been reported to be necessary to form a β-sheet structure, while a minimum length of four residues is needed for α-helical structure[26], with the latter structure in particular considered important for activity of some AMPs[26]. It would have been interesting to confirm this by characterising the structure of our peptides using circular dichroism, however this was not possible in practise due to the high levels of W, and propensity of the peptides to aggregate. As W and R are, respectively, the most hydrophobic and cationic naturally occurring residues, it seems reasonable to conclude this represents the lower size limit for the activity for all peptides composed of naturally occurring amino acids. Some studies have reported activity in shorter peptides composed of non-natural amino acids[27]. Whether this is due to more potent mechanisms of action available via non-natural amino acids, or simply more rapid catabolism of natural amino acids is unclear. However, it suggests that the conclusions of our

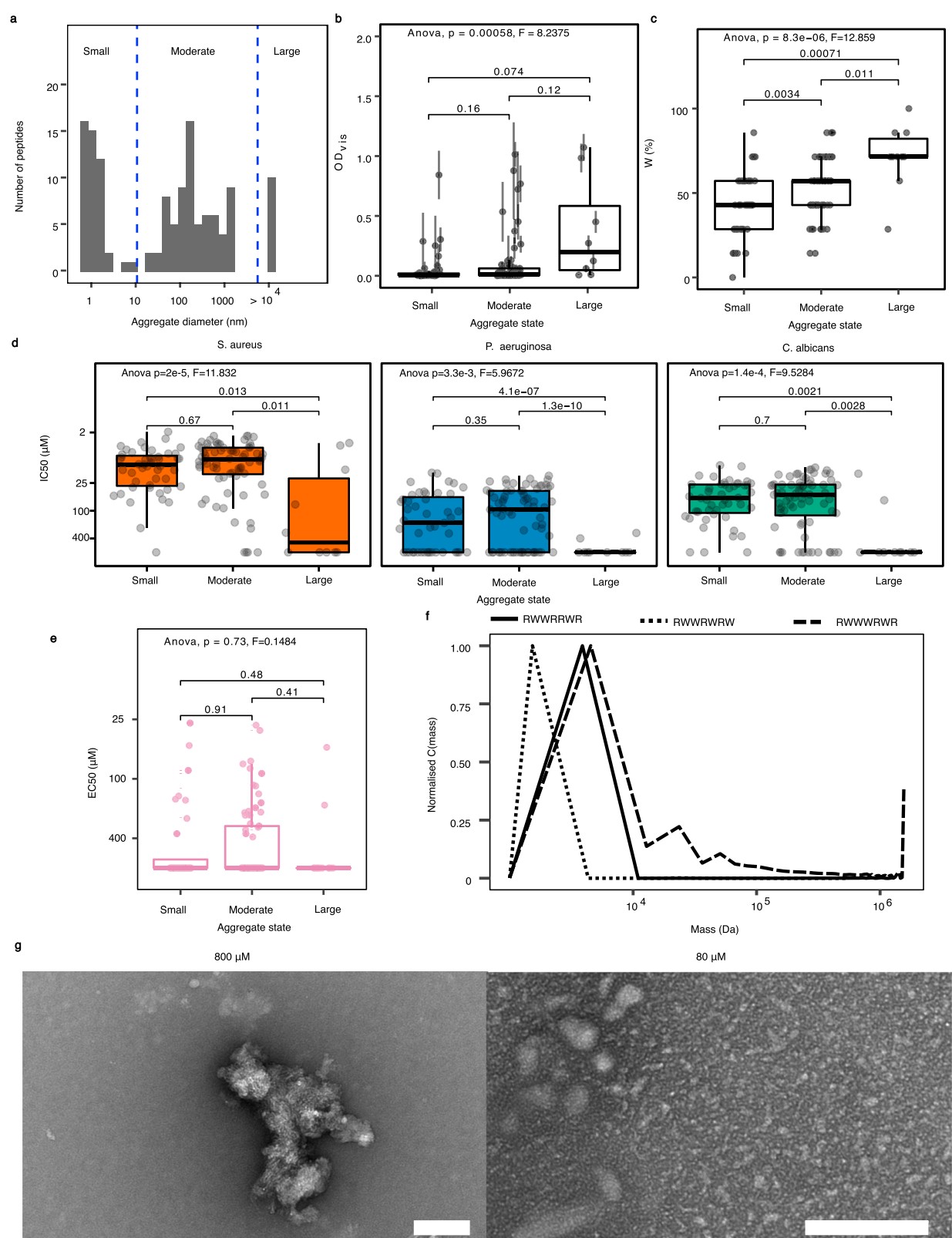

study may not extend to non-natural sequences (which are often used to stabilise clinical lead AMPs).

Our preliminary in silico analyses of our primary sequences uncovered only one protease (pepsin) which might cleave in a sequence specific manner. We found that the likely theoretical extent of pepsin digestion did correlate with activity. However, as pepsin is a gastric protease and is most active at low pH, it is unlikely to be present or active during our MIC testing. Nonetheless, more widely, propensity to aggregate might indirectly affect the accessibility of cleavage sites. In further studies it would therefore be important to confirm whether peptides within our library, in aggregation states typical at active concentrations, are differentially digested by common endogenous proteases.

**Fig. 3 Analysis of heptapeptide aggregation. a** Analysis of the size of aggregates for peptides in stock solutions (800 μM) using DLS, indicating the three size categories identified (featuring small, moderate or large aggregates; n = 47, 72 and 10, respectively). **b** Box plots indicating $OD_{vis}$ distributions (i.e. likely overall extent of aggregation) for peptides in each aggregation state. Error bars shown are ±s.e.m. (n = 4). Numbers on brackets indicate the P value from a post hoc Tukey test of the aggregate states having the same distribution. **c** Box plots showing percentage W (by number of W residues in sequence) content distributions for peptides in each aggregation state. Numbers on brackets indicate the P value from a post hoc Tukey test of the aggregate states having the same distribution. **d** Antimicrobial activity ($IC_{50}$) of peptides in each aggregation state for three test organisms. Numbers on brackets indicate the P value from a post hoc Tukey test of the aggregate states having the same distribution. **e** Haemolytic activity ($EC_{50}$) for peptides in each aggregation state. Numbers on brackets indicate the P value from a post hoc Tukey test of the aggregate states having the same distribution. (n = 254 peptides in **a**–**e**). **f** Analysis of the size of aggregates present for three different peptides found to produce substantial aggregates at 800 μM after a tenfold dilution (i.e. to 80 μM) and using AUC. **g** TEM imaging of the peptide WRWWWWW at concentrations of 80 and 800 μM. Scale bars shown are 100 nm in length.

A second general conclusion is that overall activity increases consistently with length. However, extrapolations of the relationship between activity and peptide length suggest that the specific activity peak (i.e. the length of peptide above which further additions decreases activity per residue) occurs below 20 residues, with the best estimate being at around ten residues. We previously reported variable activity in a group of cationic and hydrophobic AMPs based on apoE-derived AMPs, and a range of similar peptides we tested. Notably apoE-derived peptides (composed of the hydrophobic residues W, and the cationic residues R and K) showed reduced antibacterial activity as they were shortened from the 18mer ApoEdpL-W (WRKWRKRWWWRKWRKRWW, minimum $IC_{50}$ 7 μM in *P. aeruginosa*) to 15mers or shorter. Conversely, the heptamer peptide RWWRWWR had relatively high activity (minimum $IC_{50}$ 14 μM in *S. aureus*), despite being much shorter than inactive 9mer or 12mer truncations of ApoEdpL-W, as did the similar Y-containing octapeptide RRWYRWWR (minimum $IC_{50}$ of 21 μM in both *S. aureus* and *P. aeruginosa*)[28]. It is possible that the introduction of K residues in ApoEdpL-W may have imparted behaviour beyond that observed in peptides composed of W and R alone. However, in our current work, such disparate patterns are also present, with short active peptides and longer, less active peptides, within a wider pattern of increasing activity with length (Fig. 1a and Supplementary Movie 1). Thus our general observation that the activity increases with length does not mean that truncating a particular 18mer will not strongly impact activity for that single peptide. Calculating specific activity (i.e. activity per residue) helps us make such comparisons; while they are longer, both the octapeptide described above and ApoEdpL-W have specific activities well within the ranges explored here (126 and 168 μM, respectively, cf. Fig. 1c).

Longer peptides were associated with increases in average and maximum activity, and also in the variability of activity for a peptide of the same length. This is observed both in the increasing spread of activities in Figs. 2b and 1c, and in the increase in the roughness of the sequence-activity landscape with peptide length (Fig. 2f). This observation partly reflects an increasing maximum activity with length, while keeping the same minimum of inactivity, but also reflects the occurrence of some exceptional longer peptides with relatively high activity. However, to identify more of such peptides would require searching ever-larger sequence spaces. In this study we start to define design rules, such as the role of W clusters (Fig. 2c), on which rational design within such vast spaces may be possible. However, these rules appear incomplete as, for example, they do not explain the highlighted 65-fold difference in activity between WWWRRRR and WWWRRRW.

Alternatively, without relying on design rules, searching such spaces using iterative evolutionary approaches may be effective[29], as it is for aptamer identification, either in vitro (e.g. via the SELEX process[30]) or in silico[29]. Appropriate algorithm selection

is critical to exploring such large landscapes effectively and efficiently. For example, an algorithm starting with the best available peptide, considering variations upon it, taking the best of those and then iterating would be effective if there were a single optimal peptide and single steps away from it were progressively worse. However, this is far from the case in practice (Fig. 1a and Supplementary Movie 1), meaning that more complex algorithms would be required. Complete peptide landscapes, such as those presented here, may therefore provide a means to test and develop algorithms tailored to deal with the nuances of AMP sequence-activity relationships.

Activity was higher in peptides containing slightly more W than R, with this being the case for peptides of different length and for different microorganisms (Fig. 2b). W is the most hydrophobic naturally-occurring amino acid[31–33], due to its indole ring, a bicyclic structure comprising conjoined benzene and pyrrole rings[13]. The indole ring may enable the stable insertion of the peptide into the cellular membranes of microorganisms, as it is most stable at the interface of the hydrophilic extracellular environment and the hydrophobic interior of the cellular membrane[13]. Increasing W residues in the peptide increase membrane insertion, and also perturbation (we previously reported that W increases membrane perturbation, in apoE-derived AMPs[28]), thereby increasing efficacy. In the present study we found that heptapeptides containing four W showed the most extensive binding to anionic membranes (Fig. 4a, which are chemically more representative of bacterial membranes), with this correlating with the proportion of W found to be most efficacious (Fig. 2b), suggesting a close relationship between membrane binding and activity across these peptides consistent with previous findings for similar peptides[21,34].

Increasing peptide hydrophobicity seemed likely to be a factor causing aggregation and indeed we found that peptide aggregation increased with increasing W (Fig. 3c). We also found that peptides containing very high proportions of W showed reduced activity (Fig. 2b) as did those peptides capable of forming very large aggregates (>$10^4$ nm, Fig. 3d), with the latter being larger than the typical size of a bacterial or yeast cell. Nonetheless overall aggregation—measured by $OD_{vis}$ was slightly beneficial to antimicrobial activity, and notably our feature selection analysis (Supplementary Data Table 1) showed that the six primary structural features associated with aggregation were also found to associate with haemolytic activity ($EC_{50}$). Peptide aggregation was concentration dependent, with aggregation measured at high concentrations, so such aggregates may have partly dispersed when peptides were diluted down to the concentrations tested in the MIC experiments, although our TEM and AUC findings both confirm small aggregates are present at 80 μM, within the range of MIC experiments. Together these relationships suggest that aggregation is integral to activity until aggregate size becomes large, at which point activity diminishes due to reduced bioavailability of the active form of the molecule[35–37].

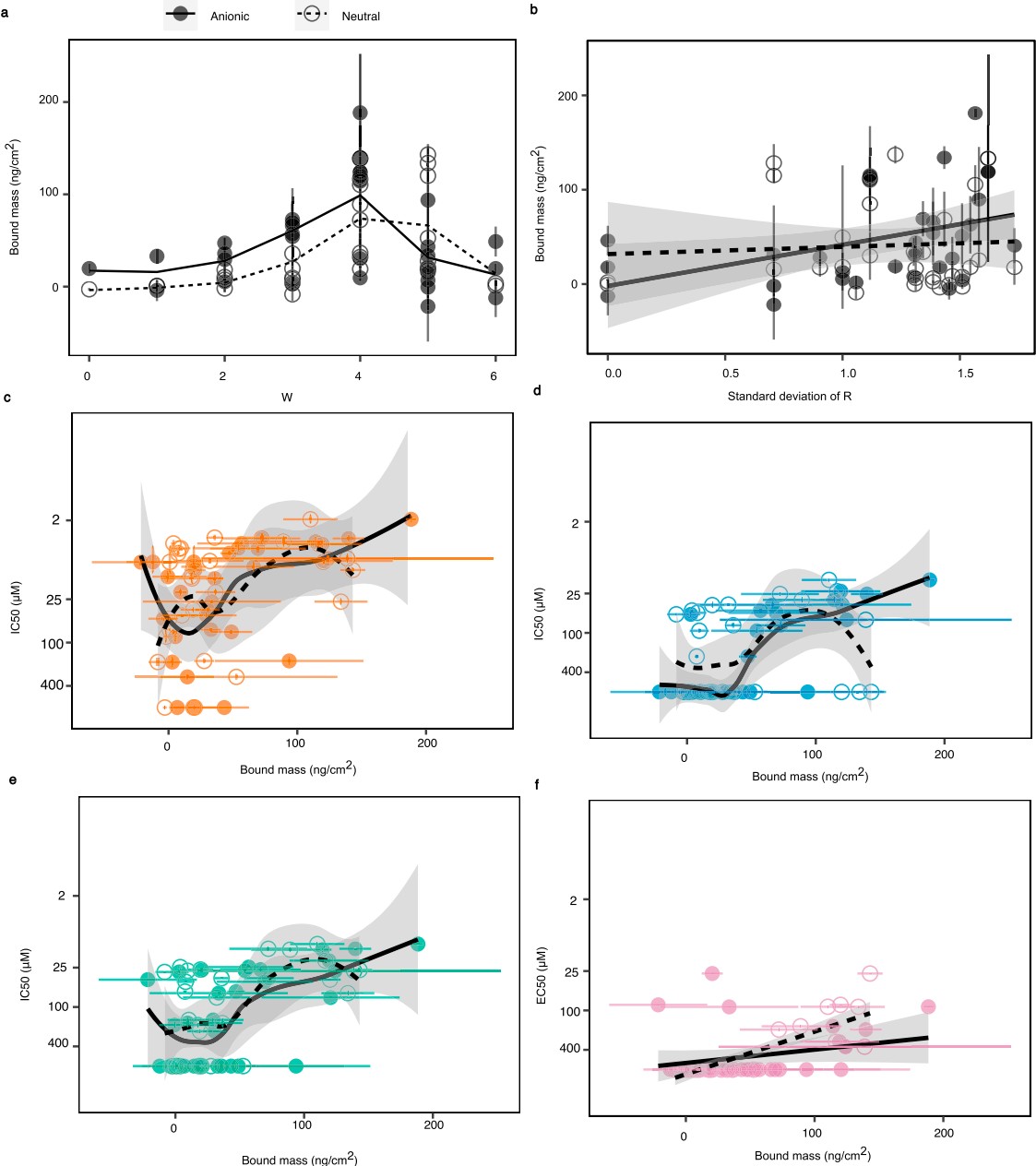

**Fig. 4 Membrane binding capacity of heptapeptides. a** Comparisons of bound mass of selected heptapeptides to anionic and neutral membranes against their W content. **b** Comparison of bound peptide mass to anionic and neutral membranes with distribution of R residues in the peptides (in the form of the positional standard deviation of R). The solid and dashed lines indicate the linear regression between bound mass and positional standard deviation of R for anionic and neutral membranes, respectively with the solid and outlined ribbons indicating the 95% confidence interval on the regression lines for the respective membranes. Analysis of the relationships between bound peptide mass to anionic and neutral membranes and antimicrobial activity against **c** *S. aureus*, **d** *P. aeruginosa* and **e** *C. albicans*. The solid and dashed lines indicate the LOESS regression between bound mass and $\log_2(IC_{50})$ deviation of R for anionic and neutral membranes, respectively with the solid and outlined ribbons indicating the 95% confidence interval on the regression lines for the respective membranes. **f** Analysis of the relationship between bound peptide mass to anionic or neutral membranes and haemolytic activity. The solid and dashed lines indicate the linear regression between bound mass and $\log_2(EC_{50})$ for anionic and neutral membranes, respectively with the solid and outlined ribbons indicating the 95% confidence interval on the regression lines for the respective membranes. All error bars shown are ±s.e.m. ($n = 4$) peptides were used throughout.

An interesting parallel has recently been reported in the AD field, where the Aβ peptide, whose misfolding and aggregation is associated with AD, has been shown to function as an AMP[20], potentially ensnaring bacteria in a 'cage-like' structure, a process dubbed bioflocculation[19,38]. In a further parallel, Aβ is thought to be most biologically active when it forms smaller aggregates (oligomers or protofibrils) with larger fibrils being relatively inert[39]. Our results suggest that AMPs, at least simple ones of the type in our complete set, also operate as small aggregates rather than individually. This may provide an additional explanation why maximal specific activity is reached for sequence of only around ten residues (individually too short to span a biological membrane), as activity is a consequence of clusters of peptides rather than peptides in monomeric form.

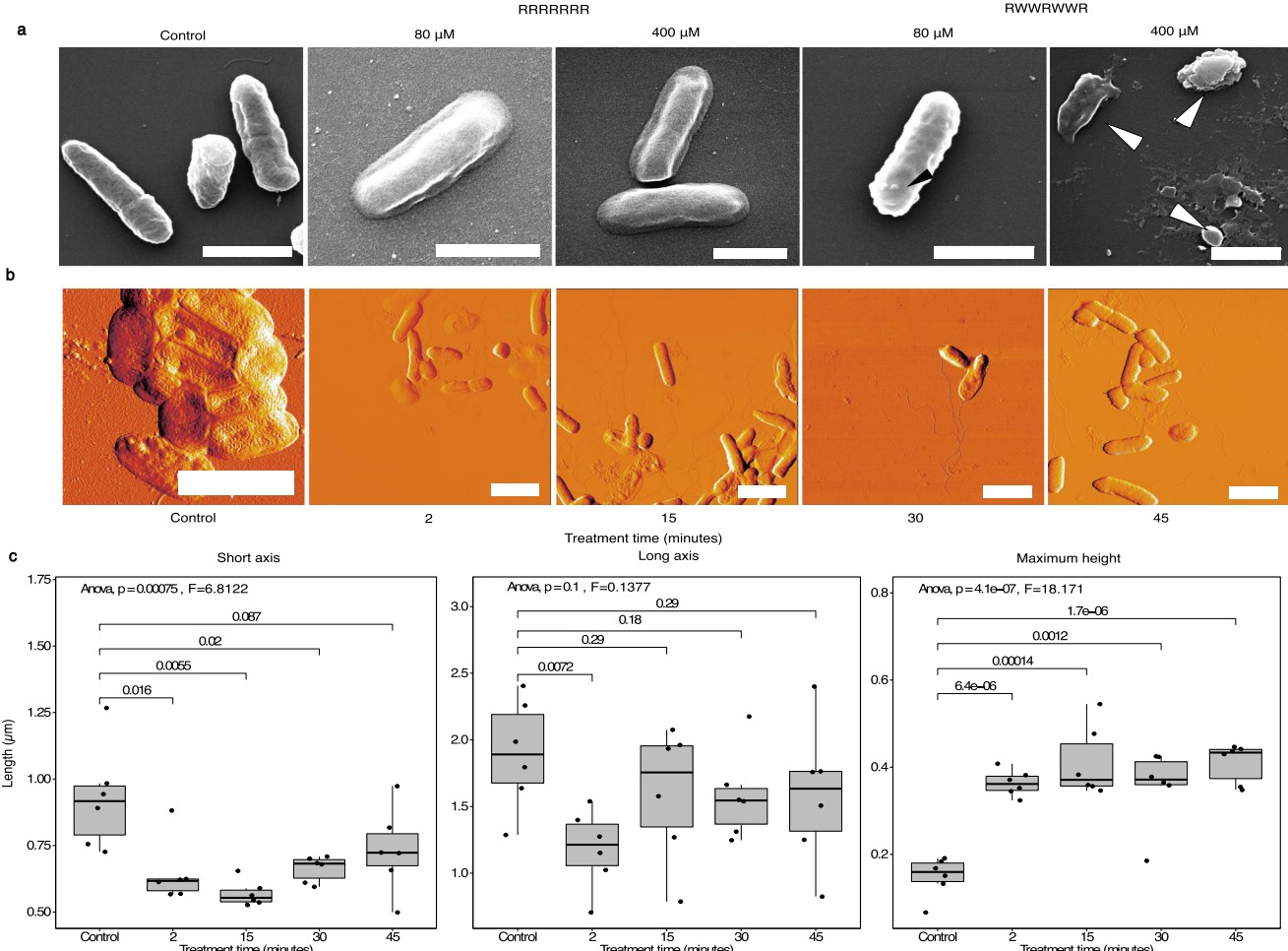

**Fig. 5 Effects of antimicrobial peptides on bacterial ultrastucture. a** Appearance of *P. aeruginosa* bacteria treated with the peptides RWWRWWR and RRRRRRR at concentrations of 80 and 400 μM, visualised by SEM. A control image of bacteria not treated with peptide is also shown. Black arrow indicates a localised protrusion (bleb) on the surface of the bacteria. White arrows indicate debris, presumably from lysed bacteria. White scalebars shown are 1 μm in length. **b** Appearance of *P. aeruginosa* bacteria treated with the peptides 80 μM RWWRWWR for various time periods, visualised by AFM. A control image of bacteria that were not treated with peptide is also shown. White scalebars shown are 2 μM in length. **c** Quantification of the average change in size of bacteria after peptide treatment for various times. Boxplots show the long axis, short axis and maximal heights of bacteria in typical AFM images from (**b**). Numbers on brackets indicate the *P* value from a post hoc Tukey test for time points where distributions either end of the bracket are significantly different. $n = 6$ cells used for analysis at each time point.

In addition to the proportion of W, the relative positioning of R and W residues also influenced the extent of antimicrobial activity (Fig. 2c); clustering of W either as doublets or as a single triplet increased activity. One explanation may be that membrane perturbation was increased locally by the presence of two or three adjacent W residues inserting at adjacent points. The increase in activity associated with W was not found for peptides containing only isolated W residues, suggesting adjacent hydrophobic residues may be necessary to deliver more substantial membrane perturbation. Our study also suggested that an even spread of R residues was important for peptide binding, though not overall activity. Even distribution of charge along the length of the peptide has previously been proposed to improve electrostatic association of AMPs with the microbial membrane attachment[34]. This is supported in our study, which revealed a significant positive association between the positional standard deviation of R (i.e. the extent to which R is distributed along the sequence) and the mass of peptide bound to the membranes observed (Fig. 4b). Notably we did not find an association between the clustering of R as doublets or triplets, and antimicrobial activity, strongly suggesting R residues

contribute through their distributed effect on membrane binding, rather than together with other adjacent R residues. R may function to promote the initial association of the peptide with the membrane, and so an even spread of R would decrease competition for binding sites between R residues, and also ensure that the full length of the peptide is held in proximity to the surface of the membrane[34]. This would thereby facilitate disruptive membrane interactions of W residues, a process which may take place after initial electrostatic association of the peptide with the membrane.

Haemolytic activity reflects the potential for an antimicrobial to have adverse activity against mammalian cells. This shows a similar relationship with tryptophan content to antimicrobial activity, though with a higher optimal content W content, close to 80% (Fig. 2e cf. Fig. 2b). This perhaps reflects the diminished role for R in mediating attraction, given the slightly positive charge reported for red blood cell membrane, which is typically +10 to 20 mV[22], compared with −20 to −50 mV found for microorganisms[40]. Consistent with this, we found that binding of heptapeptides to neutral membranes was not influenced by even distribution of R around the peptide, as had been the case for

anionic membranes (Fig. 4b), suggesting that R was less involved in mediating interactions with non-anionic membranes.

The potential importance of the bacterial cell membrane as the primary mechanism of action of these peptides was further supported by the appearance of surface changes including depressions or external protrusions after a short exposure to the peptide (Fig. 5a). As the timescales of experiments were necessarily short to test for immediate direct effects, the concentrations of peptides used were higher than those used in the MIC experiments, which had involved much longer exposure times. We therefore cannot exclude that the mode of action taking precedence after short term exposure to high concentrations is different from that after overnight incubation and antimicrobial assays. However the simplest explanation is that in both cases the changes are mediated by membrane perturbation. Protrusions have been reported previously following AMP-treatment, and were associated with loss of cytoplasmic material from the bacteria after membrane perturbation[41]. After exposure to a 400 μM concentration of active peptide, the bacteria disintegrated, perhaps due to loss of membrane integrity. Additionally, changes in both the width and height of the *P. aeruginosa* rod were observed using AFM after a very brief (2 min) peptide treatment (Fig. 5b, c). This suggests a direct effect on membrane fluidity, as treated bacteria appeared to increase in height, suggesting their membranes may be under greater pressure. This is consistent with deposition and insertion of AMPs into the membrane, potentially resulting in greater membrane 'stiffness'. This is consistent with previous studies, which have reported that other AMPs induce loss of membrane fluidity in bacteria[42].

In conclusion, we have taken an unprecedentedly comprehensive approach to explore the antimicrobial activity of short peptides, composed exclusively of arginine and tryptophan. We have determined both the presence and the limits of simple rules for defining activity across this complete sequence landscape. This involved characterising simple relationships involving length and number of hydrophobic residues and confirming how those differ for antimicrobial and haemolytic activity. Moreover, we have uncovered striking complexity even within this relatively 'simple' system, identifying large variation in activity among very closely related peptides. Some of this complexity may be caused by the two main mechanisms of activity we identified as important here: aggregation and membrane binding, which are associated with changes in cell morphology and death. Further analysis of complete sequence-activity landscapes for other amino acids opens the way for studies focused on developing novel antimicrobials fully synthetically, based on rational design. The outputs of our study (and further similar ones) may be used as a compendium of activity associated with specific shorter peptide motifs, enabling larger peptides with clinical utility to be assembled.

## Methods

**Chemicals and media.** Dimethyl sulfoxide (DMSO) was obtained from Sigma Aldrich (Poole, UK). Bacteriological growth media and phosphate buffered saline (PBS) were obtained from Oxoid (Basingstoke, UK). Defibrinated horse blood was purchased from EO Labs (Bonnybridge, UK). HEPES was obtained from MP Biomedicals (Santa Ana,USA) and NaCl was obtained from Fischer Scientific UK (Loughborough, UK).

**Peptide library synthesis.** A complete library of peptides comprising all combinations of tryptophan (W) and arginine (R) up to seven residues long was synthesised by Covalabs UK (Cambridge, UK) using 9-fluorenylmethyl-carbamate chemistry and purified using ultra performance liquid chromatography. Synthesis of two of the most hydrophobic peptides (WWWWWWR and WWWWWWW) was unsuccessful. The test library therefore comprised the amino acids R and W, 4 dipeptides, 8 tripeptides, 16 tetrapeptides, 32 pentapeptides, 64 hexapeptides and 126 heptapeptides (252 peptides in all). Peptides were manually synthesised with trifluoroacetic acid (TFA) counterions and cleavage was performed with TFA, and

C-terminal amidation and N-terminal acetylation carried out for all peptides. Wang resin was used as the synthesis substrate and peptides were purified on a C18 column. Ninhydrin was used to verify correct coupling of amino acids. Purity was verified at >80% using matrix assisted laser desorption/ionisation time-of-flight (MALDI – TOF) mass spectrometry.

**Peptide sequence analysis.** We identified a set of features to examine for each peptide sequence (referred to as the 'feature space'). Components (and the symbols used to refer to these) included the abundance of each of the amino acids (A%), the occurrence of amino acid duplets (AA) and triplets (AAA), the mean position of each amino acid ($\bar{X}_A$ and the positional standard deviation of each amino acid ($\sigma_A$) (see Supplementary Table 2 for full list). Feature selection was carried out using the Boruta wrapper algorithm (version 6.0.0[43]) in R. For all Boruta analyses, the entire feature space available was utilised. The maximum number of iterations performed by the algorithm (maxRuns) was set at 100,000 for all analysis; this value was sufficient for the algorithm to reach a decision on all features in all analyses performed. Hydrophobic moments were calculated based on the methodology outlined in Eisenberg et al. 1984[44], taken over windows of four residues, with the maximal value taken. Pepsin cleavage sites criteria were based on Keil 1992[45], with permissible cleavage sites having a W residue in the residue site before the cleavage site, so long as an R residue is not located three residue positions before the site (i.e. the motif RxW is not allowed before the cleavage site).

**Microorganism culture.** *S. aureus* ATCC 6538 and *C. albicans* ATCC 10231 were obtained from Oxoid (Basingstoke, UK), and *P. aeruginosa* PAO1 donated by the University of Nottingham. Bacteria were routinely grown using Mueller–Hinton broth or agar and yeast was grown using Sabaraud Dextrose broth or agar. Cultures were incubated aerobically on agar at 37 °C for 20 h or under the same conditions in broth with shaking at 200 rpm.

**Antimicrobial assays (IC$_{50}$ and MBC).** Planktonic cultures were obtained by inoculating single colonies from 18–20 h old agar cultures into 10 ml of sterile broth, followed by 20-h incubation. *S. aureus* and *P. aeruginosa* assays were performed using Mueller–Hinton broth and agar and *C. albicans* assays were performed using Sabaraud Dextrose broth and agar. Test organisms were isolated by centrifugation of 1 ml of the culture at 2000 × *g* for 3 min (MiniSpin®; Eppendorf, Germany), and resuspending the pellet in 1 ml of fresh sterile broth, before further dilution in sterile broth, until an optical density at 600 nm (OD$_{600}$) of 0.8 was obtained. This was diluted 1:50 in sterile broth, resulting in an inoculum of ~10$^6$ colony-forming units per ml (CFU/ml).

Peptide stock solutions of 800 μM were prepared by dissolving solid peptides (provided in pre-weighed aliquots by the supplier) in phosphate buffer solution (PBS) containing 5% (v/v) DMSO. Further dilutions of the stock solutions were prepared (at double the final test concentrations of 0.6–400 μM) in PBS in 5% (v/v) DMSO. All peptide stocks and dilutions were prepared in polypropylene centrifuge tubes. About 100 μl of the test inoculum was transferred into the wells of a polystyrene 96-well microtiter plate and 100 μl of each of the peptide concentrations were added into each test well, giving a final cell concentration of 5 × 10$^5$ CFU/ml. Each polystyrene plate was then aerobically incubated at 37 °C for 18–20 h without shaking, before measuring OD$_{600}$ using a spectrophotometer plate reader (Nanostar; BMG Labtech, Ayelsbury). A sigmoidal function was fitted to the resulting data using the following equation:

$$\text{Growth} = A + \frac{(A + B)}{1 + 2^{(Dlog_2(x) - C)}} \quad (1)$$

Where *x* is the peptide concentration, *A* and *B* are the OD$_{600}$ of the positive and negative controls respectively, *D* is a scaling factor and *C* is the IC$_{50}$. The mean IC$_{50}$ for each peptide was calculated from the results of four independent experiments, with a single technical replicate each. Specific activity was calculated by plotting IC$_{50}$ (μM) multiplied by peptide length, against peptide length.

Minimum biocidal activity (MBC) for each peptide was assessed by removing 5 μl samples from all wells in the plate after overnight growth was completed (immediately after OD$_{600}$ assessment), then adding these samples to agar and incubating them at 37 °C for 18–20 h. MBC was considered to be the lowest concentration of each peptide in which no microorganism growth was observed across three technical replicates. The mean MBC was calculated using data from four independent experiments.

**Determination of haemolytic activity (EC$_{50}$).** The haemolytic assay protocol was based on those performed in previous studies[25,46]. Erythrocytes from defibrinated horse blood were prepared by centrifugation at 2000 × *g*, 4 °C for 10 min. The pellet was resuspended in PBS and the cells washed a further two times by centrifugation (2000 × *g*, 4 °C for 10 min) and resuspension PBS a further two times. An erythrocyte test suspension was prepared by resuspending cells in PBS (20% v/v) and was stored at 4 °C, prior to use. For assays, a 1% v/v cell suspension in PBS was prepared and 100 μl aliquots transferred to polypropylene v-bottomed 96-well microtiter plates. Dilutions of the stock solution of each peptide were plated into a separate polypropylene 96-well microtiter plate (at double the final test concentrations), and 100 μl of each peptide solution was then transferred into 96-well

plate containing the test cell suspensions, before incubation at 37 °C for 1 h. The final concentration of erythrocytes in the assays were ~$5 \times 10^7$ cells/ml. Positive and negative controls were included for each row of the plate, specifically 100 µl of MilliQ water and 100 µl of PBS, respectively, each of which was added to 100 µl 1% v/v cell suspension in PBS. After incubation, non-lysed cells were removed by centrifugation at $2000 \times g$, 4 °C for 5 min. Finally, 100 µl of the supernatant was transferred to a flat bottomed 96-well plate, and the haemoglobin content of the supernatant was assessed by measured $OD_{450}$ using a spectrophotometer plate reader (Nanostar; BMG Labtech, Aylsbury, UK). Sigmodal curves based on Eq. 1, with $C$ in this case being $EC_{50}$, were fitted to the resulting data, from which the concentration of peptide resulting in 50% lysis of erythrocytes ($EC_{50}$) was determined. Mean $EC_{50}$ was calculated using data from four independent experiments, each composed of a single technical replicate.

**Stock solution turbidity ($OD_{vis}$).** Peptide stock solutions were prepared at 800 µM in PBS with 5% v/v DMSO. About 100 µl of each peptide solution was then transferred into 96-well plate and $OD_{vis}$ was then assessed by measuring $OD_{600}$ using a spectrophotometer plate reader (Nanostar; BMG Labtech, Ayelsbury). Mean $OD_{vis}$ was calculated using data from four repeat experiments, each composed of a single technical replicate.

**Dynamic light scattering (DLS).** Peptide stock solutions were prepared at 800 µM in PBS with 5% v/v DMSO. These were subjected to centrifugation at $2000 \times g$ for 5 min at 20 °C (Eppendorf 5810 R; Eppendorf, Germany) to remove dust and excessively large aggregates. 150 µl of the supernatant was transferred to a quartz cuvette and the particle size distribution at 20 °C was determined using DLS (Zetasizer NanoS; Malvern Instruments Ltd. Malvern, UK.). The number distribution (number of aggregates observed per aggregate diameter) from two independent experiments with three technical replicates each was then used to determine the mean aggregate size of each peptide across the 12 replicates. Aggregates were then categorised into three classes, those producing small aggregates (<10 nm diameter), moderate aggregates (diameter between 10 and $10^4$ nm) or large aggregates (>$10^4$ nm diameter).

**Analytical ultracentrifugation.** Peptide solutions were prepared at 800 µM in PBS with 5% v/v DMSO, along with subsequent dilution in the same diluent at 80 µM. Sedimentation equilibrium experiments were performed at 20 °C using an analytical ultracentrifuge (ProteomLab XL-A; Beckman Coulter. Brea, USA). Double carbon-epoxy cells were used to house the sample and reference buffer, and the samples were subjected to centrifugation at $50,000 \times g$ for 14 h. The absorbance across the cell was measured at 280 nm at regular intervals during the run to ascertain when equilibrium had occurred. Molecular masses at equilibrium were determined as a single ideal species using the software Heteroanalysis[47]. Data shown is from single independent experiments.

**Lipid vesicle preparation.** Phospholipid 1,2-dioleoyl-sn-glycero-phosphocholin (DOPC) and 1,2-dioleoyl-sn-glycero-3-phospho-rac-(1-glycerol) (DOPG) were obtained from Avanti Polar Lipids Inc (Alabama, USA). Neutral lipid bilayers were formed from vesicles composed entirely of DOPC, whilst negatively charged bilayers were composed of DOPC and DOPG at a ratio of 4:1 respectively. Lipids were dissolved in chloroform in a glass vial and then dried to a film under nitrogen gas. The lipids were then resuspended in HEPES buffer with 150 mM NaCl at 1 mg/ml and frozen at −80 °C for 30 min. This suspension was thawed, before sonication at 10 Hz in 6 to 10 bursts, each lasting 1 min (MSE Soniprep 150 plus; MSE Centrifuges Heathfield, UK). The size of the resulting vesicles was assessed using DLS (with a 90° angle) to verify the vesicles were of a target diameter of 30 nm (Zetasizer NanoS; Malvern Instruments Ltd. Malvern, UK.) and the suspension was stored in 200 µl aliquots 4 °C and used for experiments within 2 weeks of preparation. Vesicle size was reassessed DLS before each use to ensure that they were of a target diameter of 30 nm.

**Membrane binding analysis.** Membrane-peptide interactions were assessed using a QCMD monitoring (Q-Sense Omega; Biolin Scientific. Manchester, UK); a subset of heptapeptides ($n = 29$) was analysed, representing the 20 pairs of peptides that had the largest differences in $IC_{50}$ values whilst differing by only a single substitution in the peptide sequence. The stock vesicle suspension was diluted 1 in 5 with HEPES, and then 10 µl of 1 M $CaCl_2$ was added. $SiO_2$ sensors were cleaned using UV/Ozone (Bioforce Nano Procleaner) and lipid bilayers were prepared by depositing 0.2 mg/ml vesicle suspension at a rate of 25 µl/min for 5 min, followed by removal of excess lipid by adding HEPES buffer onto the chip at 25 µl/min for 5 min. The frequency shift of the quartz chip was monitored throughout to ensure that the lipid bilayer had formed correctly with a frequency of −25 Hz. Once a stable bilayer had been formed, peptide solutions at 20 µM (chosen due to its location centrally within the $IC_{50}$ distributions of the peptides) in HEPES buffer were deposited onto the bilayer at a rate of 20 µl/min for 9 min. Next, HEPES buffer was added at a rate of 25 µl/min for 5 min, to remove any weakly bound or unbound peptide. All parts of the experiment were performed at 20 °C. Frequency and energy dissipation changes were recorded using Q-soft and the frequency shift in the fifth octave was converted to mass shift using Q-Tools software (Q-sense

Omega, BiolinScientific/Q-Sense, Sweden). The mass change between the stable bilayer and the end of the final washing step was then used to determine the amount of peptide bound to the membrane. Four independent replicates were performed and bound mass values shown are means over these replicates. Errors shown are standard errors on the mean over all replicates. The R package changepoint (version 2.2.2[48]) was used to determine the average mass of peptide bound to the bilayer at each step.

**Transmission electron microscopy.** Peptides were prepared at 800 µM in PBS with 5% v/v DMSO, and any excessively large aggregates or dust removed by centrifugation ($2000 \times g$, 5 min). Further dilutions (80 µM) were prepared in PBS with 5% v/v DMSO, and the latter diluent was also used to prepare control (peptide-free) grids. Carbon film 200 mesh copper transmission electron microscope (TEM) grids (Agar Scientific Ltd. Stanstead, UK) were initially functionalized using a glow discharger (K100X; Quorum technologies Ltd. Laughton, UK) under a current of 25 mA for 30 sec. These were then placed onto 10 µl droplets of each test sample solution for 1 min, and then transferred onto 10 µl droplets of 1% w/v uranyl acetate for a further minute. Excess solution was removed using filter paper and sample grids were air dried for at least 1 h. Grids were imaged using a Tecnai™ T12 BioTwin TEM (FEI Company. Hillsboro, USA). Images were cropped and scale bars were added using ImageJ. Images are representative of single independent experiments.

**Scanning electron microscopy (SEM).** Single *P. aeruginosa* colonies were isolated from agar cultures and added to 10 ml of broth, before incubation for 18–20 h at 37 °C. Cells were then harvested by centrifugation of 1 ml of the broth culture at $2000 \times g$ for 5 min (MiniSpin®; Eppendorf. Hamburg, Germany), and the resulting pellet suspended in PBS. This was repeated twice to wash cells, before diluting to prepare a microbial suspension of ~$10^7$ CFU/ml in PBS (based on measurement of $OD_{600}$). This was treated with test peptides (RRRRRRR or RWWRWWR), by adding 200 µl of peptide (at 160 or 800 µM) to 200 µl of the bacterial inoculum, resulting in final peptide concentration of 80 and 400 µM. The mixtures were then incubated at 37 °C for 1 h. Controls were also prepared in parallel, in which bacteria were treated with 200 µl PBS containing 5% v/v DMSO.

After incubation, 100 µl of each sample was deposited on to poly-L-lysine treated mica sheets (Agar Scientific Ltd; Stansted, UK). Samples were fixed by applying 2.5% w/v gluteraldehyde, 4% w/v paraformaldehyde in 100 mM HEPES to each mica sheet and incubating overnight at 4 °C. Mica sheets were washed in MilliQ water and dehydrated through 10 min exposures to ethanol at 50, 60, 70, 90 and 100% ethanol (latter wash was carried out twice). Critical point drying (CPD) was then performed (K850; Quorum Technologies Ltd. Laughton, UK). The sample was sputter coated with Au under argon gas using a sputter coater (SC7620 Mini sputter coater; Quorum Technologies Ltd. Laughton, UK) for a period of 75 sec under a current of 18 mA. Samples were imaged by SEM using a Quanta™ FEG 250 scanning electron microscope (FEI Company, Hillsboro, USA). Images were cropped and scale bars were added using ImageJ. Images are representative of three independent experiments.

**Atomic force microscopy (AFM).** A suspension of *P. aeruginosa* at ~$10^7$ CFU/ml in PBS was prepared as for SEM; 500 µl of this suspension was incubated with 500 µl of RWWRWWR peptide (200 µM in PBS; 5% v/v DMSO) at 37 °C with shaking at 200 rpm for 45 min. As a negative control, 500 µl bacterial suspension was incubated with PBS containing 5% v/v DMSO. Samples of 50 µl were withdrawn from the suspensions at 2, 15 and 45 min for imaging by AFM.

Chips were prepared by adsorbing 50 µl of poly-L-lysine onto a cleaved mica chip for 3 min. Excess poly-L-lysine was washed off by adding $10 \times 200$ µl aliquots of MilliQ water and excess water was removed using filter paper. Test samples were deposited onto the chips and left to adhere for 1–2 min, before further washing using $10 \times 200$ µl aliquots of MilliQ water, blotting off excess with filter paper, before leaving chips to air-dry overnight.

Chips were scanned using a Bruker multimode 8 (Bruker, Billerica, USA) in ScanAsyst air mode. All images were taken at a resolution of 512 points per dimension unless stated otherwise, and are representative of single technical experiments. Images obtained were processed and different length measurements were performed using Gwyddion version 2.48 (http://gwyddion.net/).
Measurements were taken of the long axis length, short axis length and maximum vertical height of the bacterium above the chip surface. Measurements were taken over six different bacteria ($n = 6$).

**Statistics and reproducibility.** R software (version 3.5.0) was used for the majority of analyses performed. Graphs were plotted using the ggplot2 and ggpubr packages in R (versions 3.3.2 and 0.4.0 respectively). Two-dimensional LOESS smoothing was carried out using the geom_smooth function in the ggplot2 package. Unless otherwise stated, all errors reported are ±s.e.m. The function used to calculate Spearman's Rho was the spearman.test function from the pspearman R package (version 0.3-0), and the AS89 approximation algorithm used in this analysis. Tukey tests and one-way ANOVA analyses were performed using the stats package in R. All statistical testing performed was two-sided. The statistical model shown in Fig. 1c was created by regressing the 90% quantile of the logged $IC_{50}$ value against peptide length for all combinations of length and organism where there were any

active peptides ($n = 240$, 224 and 224 peptides for *S. aureus*, *P. aeruginosa* and *C. albicans*, respectively). A logistic curve was used, estimating a common inactive intercept among organisms and the remaining three parameters (corresponding to active intercept, midpoint and slope) free to vary among organisms. The credibility intervals for the maximum specific activity against each organism were estimated using quantiles of 8000 draws from the model's posterior distribution. Variance was allowed to increase with length as this was found to improve the model (difference in LOOIC = $3.7 \pm 3.0$). The model was fitted using the *brms* package (version 2.13.0[49]) using the asym_laplace family and four MCMC chains, each with 2000 iterations of burn-in, checking convergence numerically (Rhat = 1.00 throughout) and visually. Priors were taken from a similar logistic model, not accounting for any variation among organisms, fitted using the gnls function in the nlme package (version 3.1-137[50]). A bivariate model of $OD_{vis}$ and the logarithm of $IC_{50}$ as logistic functions of length ($n = 254$ peptides), was fit in the same way except using robust regression (the 'student' family) and including a residual correlation between the variables. For all boxplots shown, the centre line in the box indicates the median of the data, the interquartile range is indicated by the box with the edges located at Q1 and Q3, and the whiskers of the plot indicate Q1-1.5xIQR and Q3+1.5xIQR.

For the antimicrobial and haemolytic assays four biological replicates were used ($n = 4$). In the case of the antimicrobial assays biological replicates were taken as different cultures from stock, and for the haemolytic activities, different batches from supplier were taken as biological replicates. For inhibitory and haemolytic assays a single technical replicate was performed for each biological replicate, however for microbiocidal assays three technical replicates were performed. For DLS analysis two independent experiments were performed ($n = 2$), AUC was performed using a single independent experiment ($n = 1$), whilst for stock solution turbidity analysis and membrane binding analysis four independent experiments were performed ($n = 4$). Independent experiments in these cases were taken as these performed with different syntheses of the peptides. For stock solution turbidity analysis, AUC and membrane binding analysis single technical replicates were performed for each independent experiment, whilst for DLS analysis three technical replicates were performed for each independent experiment. For the SEM analysis three biological replicates were implemented ($n = 3$) whilst for AFM one biological replicate was used ($n = 1$). As with the antimicrobial assays, biological replicates were taken as being different cultures generated from stock. A single technical replicate was performed for each biological replicate in both SEM and AFM experiments.

**Landscape analysis**. For the antimicrobial and haemolytic activities (reciprocal $IC_{50}$ ($1/IC_{50}$), MBC ($1/MBC$) and $EC_{50}$ ($1/EC_{50}$)) were used as the landscape variables. Peak fraction was determined as the fraction of peptides within the landscape where activity was greater than the nearest neighbours ($A_i > A_{nn}$). $A_{nn}$. Neighbours were calculated using the raw longest common substring distances (defined as the number of deletions in both sequences required for the sequences to be equal). Deviation from additivity was defined as the sum of residuals squared for a linear model fit through the landscapes divided by the average slope of the linear model, which is often referred to as the roughness over slope or *r/s*. The peak fraction and *r/s* parameter are based on work by Szendro et al.[51]. Network plots were created by connecting sequences at an edit distance (restricted Damerau–Levenshtein distance) of 1, and laid out in three dimensions using a Fruchterman and Reingold algorithm using the sna and rgl packages in R. Distances were calculated using the stringdist R package (version 0.9.5.5).

**Reporting Summary**. Further information on research design is available in the Nature Research Reporting Summary linked to this article.

## Data availability
The raw data and R syntax to reproduce all the analyses, figures, tables and supplementary tables in the published article are available as collection on Figshare (https://doi.org/10.6084/m9.figshare.c.5104931)[52]. All other data are available from the corresponding author on reasonable request.

## Code availability
The R syntax to reproduce all the analyses, figures, tables and supplementary tables in the published article are available as a collection in Figshare (https://doi.org/10.6084/m9.figshare.c.5104931)[52]. All other code is available from the corresponding author on reasonable request.

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

## Acknowledgements

The authors acknowledge the BBSRC Doctoral training partnership (BB/M011208/1), the family of Sir Kenneth Murray and the University of Manchester for support for SC. We would also like to thank Professor Douglas Kell (University of Liverpool) for connecting the authors and initiating our collaboration, the staff of the BioAFM, biomolecular analysis and electron microscopy facilities of the University of Manchester's Faculty of Biology, Medicine and Health for technical support and Pippa Knight for editing the Supplementary Movie 1.

## Author contributions

S.C. carried out experimental work, assisted by T.A.J. in design and execution of the AUC and membrane binding studies. S.C., L.K.H., C.G.K. and C.B.D. conceived the study, and contributed to the design of experiments, interpretation of the results, wrote and edited the manuscript.

## Competing interests

The authors declare no competing interests.
