## [Peer Review File · Communications Biology]

Reviewers' comments:

Reviewer #1 (Remarks to the Author):

In this manuscript, de novo peptides comprised exclusively of Trp and Arg residues, in all possible combinations within the limit of a 7 amino acid sequence, were designed and tested for antimicrobial activity. Using this complete set of 254 constructs, the authors aimed to identify patterns relating primary sequence to antimicrobial potency towards building an "activity landscape" for rationale AMP design. From their work, it was observed that the disruption of gram-positive and gram-negative bacteria, yeast, and mammalian cells all correlated positively with peptide length, whereby a sequence containing 4-5 residues presented as the minimum length required for measurable activity. It was determined that sequences hosting slightly more Trp than Arg residues conferred enhanced functionality at microbial interfaces, presumably due to increases in hydrophobicity, giving rise to greater affinity for bilayer engagement. Similarly, the presence of 2-3 consecutive Trp residues was also shown to mediate improved antibacterial efficacy. Lastly, the authors comment on the potential benefits of peptide aggregation as a mode of action for the experimental cohort, in which small peptide oligomers are thought to entrap microbial cells in a cage-like manner.

This manuscript adopts an innovative approach towards building a comprehensive database for de novo antimicrobial peptide design. The paper was well-written and strengthened by elegantly explored experiments. While the exclusivity of this work to sequences harbouring only Trp and Arg residues limits the broad application of the current findings, it provides a compelling branching point for future studies aimed at a systematic understanding of both synthetic and natural antimicrobial peptide design. In preparing a final version of this paper for publication, the authors should consider the following comments and suggestions:

- A major limitation barring the application of peptide therapies pertains to their low in vivo bioavailability—this presents as a fundamental consideration when building a database for pharmaceutical peptides based on natural amino acids. Has it been considered to run a series of digestion assays to map out any potential trends in the primary sequence that can help to evade degradation?
- From the gathered results, hemolytic activity was not shown to be correlated with aggregation (Line 220, Fig. 3d). This was interesting as one might assume aggregation to impact the affinity of the peptides for mammalian membranes (i.e., shielding of hydrophobic regions leading to reduced hydrophobic opportunities with the bilayer). Would it be possible to expand on this finding by analyzing the primary sequences? For instance, providing rationale on why certain aggregates were more conducive to hemolytic damage while others did not affect this parameter. This may be useful in deducing patterns for peptide aggregates that offer enhanced activity at bacterial interfaces while hosting a protective function against human cell disruption.
- In this work, membrane-peptide binding was considered for a subset of 32 peptides. Are there circular dichroism spectra to complement this set of data? As helical affinity is known to influence antimicrobial activity in natural AMPs, it would be interesting to assess whether binding, secondary structure, and activity are also closely related within the context of these shortened constructs. This would also provide insight on the possible mechanisms adopted by the various sequences.
- With respect to the morphological studies, *P. aeruginosa* was exposed to 80µM-400µM of the active RWRWWR peptide. Given that RWRWWR has a much lower IC50, is there a reason why a reduced concentration was not used as a more direct assessment of membrane damage in response to peptide treatment?
- Several aspects of the presented findings seem to hint at the potential importance of amphipathicity towards antimicrobial potency (for instance, the distribution patterns for Arg with respect to anionic membrane binding). Has it been considered to graph this relationship to include as part of the supplementary documents?
- Given the extensive number of findings in this manuscript, it may be worthwhile to report on the

therapeutic index for each peptide as part of the supplementary document (ex. as a concentric ring system similar to those presented in Fig.1). This may allow readers to more readily orient towards the most pharmacologically relevant design parameters.

- Figure 1, the ring chart system may benefit from enlargement. As is, the outermost residues are not legible for the smaller diagrams.
- Line 261, potential typo: heptapeptide not pentapeptide?
- Line 761, potential mislabelling: figure legend should include label "(b)" before "Box plots indicating..."

Reviewer #2 (Remarks to the Author):

The researchers conducted a robust and unprecedented work. They systematically studied the possible combinations of tryptophan (W) and arginine (R) up to a length of 7 residues in order to carry out a meticulous and detailed study of the quantitative structure-biological activity relationship. Furthermore, this work will provide valuable insights for the design of short peptides with increased biological activity. However, some comments should be considered.

The researchers conducted robust and unprecedented work. They systematically studied the possible combinations of tryptophan (W) and arginine (R) up to a length of 7 residues in order to carry out a meticulous and detailed study of the quantitative structure-biological activity relationship. Furthermore, this work will provide valuable insights for the design of short peptides with increased biological activity. However, some comments should be considered:

Line 100: A previous research (J. Med. Chem.2003, 46, 9, 1567–1570) also studied possible combinations of arginine and tryptophan in relation to antimicrobial activity.

Line 123: Fig 1a, d, e should improve image resolution. Line 147: Should be included in methodology.

Lines 218-220: The influence of the aggregation states of peptides on the different microbial species is poorly addressed in the discussion.

Lines 238-244: In the methodology, a titration process is described, but the results show only the mass change after the addition of the entire volume of peptide. Some peptides could occupy all binding sites at lower concentrations than others affecting affinity and other correlations with biological activity. The peptide-lipid molar ratio after each titration and K affinity value could also be reported. Large standard deviations are observed in Figure 4. The number of replicates in the methodology is not clear.

Line 259: Why was this strain used? *S. aureus* showed greater sensitivity to peptides. Line 265: How do you confirm that a microorganism is viable with an image?

Line 277: The discussion is well oriented. However, the structure-antimicrobial activity relationship on the different microorganisms (Gram-negative, Gram-positive and fungus) is poorly addressed.

Line 365: 80uM?

Line 397: This could be supported by the K value.

Line 421: If it is less compressed, it should be less rigid.

Line 456: The process should be more detailed due to the amount of peptides synthesized. Is it automated synthesis? (reference of the device used). If it was manual synthesis, which resin was used? controls for each coupling? Which column was used for purification?

Line 463: cleaved with TFA?

Line 490: ¿what kind of broth?

Lines 493-495: OD600 of 0.8 can be equivalent to a greater number of bacteria than that reported since only an OD600 of 0.1 is equivalent to approximately 1×10^8 CFU / ml [1].

Line 503: This culture conditions were used for *C. albicans*?

Line 506: The IC50 would not be a good indicator to start with the MBC tests due to the variability of this indicator. After 20 hours of incubation, it is very likely that 50% of live bacteria will grow exponentially. Furthermore, the number of CFU and IC50 can vary significantly in an interval of 2 hours (18-20h) since the bacteria duplicate every 30 minutes, providing very imprecise results. Bacterial growth time should be strictly controlled. Report the Minimum Inhibitory Concentration (MIC) is suggested since the MIC is widely accepted and is considered the "gold standard" for determining the susceptibility of organism to antimicrobials and is therefore used to judge the performance of all other methods of susceptibility testing [2]. In addition, the MIC is listed as the starting point for larger pre-clinical evaluations of novel antimicrobial agents [3]. Follow the CLSI guide would be appropriate [4].

Line 521: How many erythrocytes would be undergoing the treatment? Line 531: ¿Positive control used? Equation employed for calculate EC50? Line 547: Mention the dual angle used.

Line 571: What was the temperature used for the preparation?

Line 577: Small unilamellar vesicles SUVs are characterized by being very unstable and tend to aggregate to form MLVs over time. Were size characterizations performed before being used in biophysical techniques?

Line 616: Why was this bacterium chosen? Line 977: check reference 45. Including in supplementary material a multivariate analysis between charge / amphipathicity and antimicrobial / hemolytic activity would be very interesting.

REFERENCES

1. Wiegand, I.; Hilpert, K.; Hancock, R.E. Agar and broth dilution methods to determine the minimal inhibitory concentration (MIC) of antimicrobial substances. *Nat Protoc* 2008, 3, 163–175.
2. Andrews, J.M. Determination of minimum inhibitory concentrations. *J Antimicrob Chemother* 2001, 48 Suppl 1, 5–16.
3. O'Neill, A.J.; Chopra, I. Preclinical evaluation of novel antibacterial agents by microbiological and molecular techniques. *Expert Opin. Investig. Drugs* 2004, 13, 1045– 1063.
4. CLSI M45. Methods for Antimicrobial Dilution and Disk Susceptibility Testing of Infrequently Isolated or Fastidious Bacteria ; Proposed Guideline; 2015; ISBN 1562385836.

Reviewer #3 (Remarks to the Author):

The current manuscript 'The Lexicon of Antimicrobial Peptides: a Complete Set of Arginine and Tryptophan Sequences' by Clark et al tested antimicrobial and hemolytic functions a complete set of short peptides, composed of arginine and/or tryptophan residues. Several similar works have been reported as cited in the manuscript. One of very important paper (Zhigang Liu, Anna Brady, Anne Young, Brian Rasimick, Kang Chen, Chunhui Zhou, Neville R Kallenbach, *Antimicrob Agents Chemother.* 2007;51(2):597-603. Length effects in antimicrobial peptides of the (RW)_n series) also reported similar idea to design and screen optimal antimicrobial peptides, but the current manuscript does not cite the paper. In the paper of 2007, they have found the shortest antimicrobial peptides with the length of two residues. Given some previous reports, I do not think that the current work has provided novel findings and new advances.

Reviewer 1.

We were delighted that the reviewer commented on our innovative approach, and believed that our paper was well-written and strengthened by elegantly explored experiments, and provides a compelling branching point for future studies.

1. Bioavailability.

A major limitation barring the application of peptide therapies pertains to their low *in vivo* bioavailability—this presents as a fundamental consideration when building a database for pharmaceutical peptides based on natural amino acids. Has it been considered to run a series of digestion assays to map out any potential trends in the primary sequence that can help to evade degradation?

The reviewer raised the valid point that peptide biological stability is an important consideration in developing peptide-derived drugs. However, as this depends on the diversity of the primary sequences of the peptides, and as our peptides were comprised exclusively of W or R, it is unlikely that the biological stability of our peptides would be highly variable. Nonetheless, prompted by the reviewer's suggestion, we have carried out *in silico* analysis of this, and did find that pepsin would cleave our peptides in a variable manner, as its ability to cleave at W depends on the position of R within the sequence. We have included an additional figure (Ext Data 3a), which shows that the propensity of the peptides to be cleaved by pepsin does correlate positively with antimicrobial activity, although this enzyme was unlikely to be present in our assay system. We have also provided additional narrative in the results and discussion sections on this topic.

2. Haemolytic Activity and Aggregation.

From the gathered results, hemolytic activity was not shown to be correlated with aggregation (Line 220, Fig. 3d). This was interesting as one might assume aggregation to impact the affinity of the peptides for mammalian membranes (i.e., shielding of hydrophobic regions leading to reduced hydrophobic opportunities with the bilayer). Would it be possible to expand on this finding by analyzing the primary sequences? For instance, providing rationale on why certain aggregates were more conducive to hemolytic damage while others did not affect this parameter.

This may be useful in deducing patterns for peptide aggregates that offer enhanced activity at bacterial interfaces while hosting a protective function against human cell disruption.

The reviewer raised an interesting topic, questioning whether primary sequences could be examined to test whether certain sequences resulted in forms of aggregation, which were more likely to increase hemolytic activity. In fact this point is addressed by our Boruta analysis, (Extended Table 2) which shows that six specific primary structural features associated with aggregation (as indicated by OD_{vis}) were also identified as primary structural features found to associate with EC_{50} . We have added a note to the discussion to draw attention to this point.

3. Membrane binding / CD/ helicity.

In this work, membrane-peptide binding was considered for a subset of 32 peptides. Are there circular dichroism spectra to complement this set of data? As helical affinity is known to influence antimicrobial activity in natural AMPs, it would be interesting to assess whether binding, secondary structure, and activity are also closely related within the context of these shortened constructs. This would also provide insight on the possible mechanisms adopted by the various sequences.

We did attempt to carry out circular dichroism (CD) analyses of the peptides, however CD is difficult for sequences containing W, and was also limited by the presence of peptide aggregates, and so unfortunately it was not possible to obtain reliable CD data with our library. We have added a note to the discussion to confirm this point. Given our peptides are short, stable alpha-helical formation may not be very common and random coil orientation seems as likely.

4. Peptide concentration for morphological studies

With respect to the morphological studies, *P. aeruginosa* was exposed to 80µM-400µM of the active RWWRWWR peptide. Given that RWWRWWR has a much lower IC50, is there a reason why a reduced concentration was not used as a more direct assessment of membrane damage in response to peptide treatment?

This is a useful point which we have not properly addressed in our manuscript. A key aim of the morphological studies was to determine whether any effect was apparent following a brief exposure, as this would suggest peptides were able to damage the membrane (which would be exposed immediately to the peptide) rather than needing to be taken into the cell to enable action through a more complex intracellular mechanism. The antimicrobial activity seen at lower concentrations in our MIC experiments occurred after a 15 hour incubation. It was therefore necessary to increase the treatment concentration for morphological studies to ensure any effect would be apparent over these much shorter timescales (2-60 min, rather than 15hr). We have included a note in the discussion on this point.

5. Amphipathicity

Several aspects of the presented findings seem to hint at the potential importance of amphipathicity towards antimicrobial potency (for instance, the distribution patterns for Arg with respect to anionic membrane binding). Has it been considered to graph this relationship to include as part of the supplementary documents?

This was an interesting suggestion, and we have therefore added an additional extended data figure in which we have plotted hydrophobic moment against activity and found a weak correlation (Extended Data Figure 3b).

6. Therapeutic Index

Given the extensive number of findings in this manuscript, it may be worthwhile to report on the therapeutic index for each peptide as part of the supplementary document (ex. as a concentric ring system similar to those presented in Fig.1). This may allow readers to more readily orient towards the most pharmacologically relevant design parameters.

This was an interesting suggestion, and we have therefore added additional extended data panels in which we have calculated the therapeutic index of the peptides and arranged these using the concentric ring system (Extended Data Fig. 1e), and also plotted these relative to W content of the peptides (Extended Data Fig. 2c).

Minor Changes:

1. Figure 1, the ring chart system may benefit from enlargement. As is, the outermost residues are not legible for the smaller diagrams.

We have made the smaller ring charts slightly larger, as suggested. The outermost residues in the smaller diagrams are legible in four versions of the Figure provided at high resolution. Additionally

the identity of the residues in the outermost ring can also be easily identified using the larger diagram as a key.

2. Line 261, potential typo: heptapeptide not pentapeptide?

We thank the reviewer for spotting this typo: we have now changed this to heptapeptide.

3. Line 761, potential mislabelling: figure legend should include label "(b)" before "Box plots indicating..."

We thank the reviewer for spotting this typo: we have now added the label (b).

Reviewer 2.

We were very pleased that the reviewer commented our approach was innovative, and believed that our study to be meticulous and detailed, and that it will provide valuable insights for future design of short peptides. We thank the reviewer for the very thorough and detailed range of comments and suggestions provided.

1. Line 100: A previous research (J. Med. Chem.2003, 46, 9, 1567–1570) also studied possible combinations of arginine and tryptophan in relation to antimicrobial activity.

We thank the reviewer for highlighting the 2003 article by Strom et al., which we had cited. This paper measured the antibacterial activity in just 12 of the 256 peptides which are found in our library. Its focus was on measuring activity in additional non-natural derivatives of the shortest of these peptides (in particular non-natural dipeptides). The Strom paper did not systematically combine R and W in all possible permutations as we have done, instead selecting the 12 peptides common to our library in a non-systematic manner. We have added the phrase 'in most or all possible permutations' to this line of the text to make our point more clearly.

2. Line 123: Fig 1a, d, e should improve image resolution.

The final submitted files contained images of much higher resolution.

3. Line 147: Should be included in methodology.

We have moved this line to the appropriate section of the Methods section.

4. Lines 218-220: The influence of the aggregation states of peptides on the different microbial species is poorly addressed in the discussion.

We have added a note to confirm that the aggregation states were similar for all three organisms.

5. Lines 238-244: In the methodology, a titration process is described, but the results show only the mass change after the addition of the entire volume of peptide. Some peptides could occupy all binding sites at lower concentrations than others affecting affinity and other correlations with biological activity. The peptide-lipid molar ratio after each titration and K affinity value could also be reported. Large standard deviations are observed in Figure 4. The number of replicates in the methodology is not clear.

We were not able to assess the peptide-lipid molar ratio or K affinity values, but have now added information on replicate numbers to the Methods section.

6. Line 259: Why was this strain used? *S. aureus* showed greater sensitivity to peptides.

We used *P aeruginosa* in these experiments as its larger overall size and rod shape might enable more subtle morphological changes induced by the peptides to be visualized, compared to the smaller spherical *S. aureus*. Additionally in preliminary experiments we found that *S. aureus*, when used under the conditions required for the electron microscopy experiments, grew in aggregates which made interpretation of the images more difficult.

7. Line 265: How do you confirm that a microorganism is viable with an image?

To avoid any confusion we have changed the word 'viable' to 'intact' in the text.

8. Line 277: The discussion is well oriented. However, the structure-antimicrobial activity

relationship on the different microorganisms (Gram-negative, Gram-positive and fungus) is poorly addressed.

We thank the reviewer for spotting this omission. We have now added some discussion on this point to the first page of the discussion section.

9. Line 365: 80uM?

We have changed uM to μ M.

10. Line 397: This could be supported by the K value.

We were not able to assess the K affinity values.

11. Line 421: If it is less compressed, it should be less rigid.

We thank the reviewer for pointing out that these lines as written were inconsistent. We have rephrased this section as we did not intend to give the impression that the bacteria would be less compressed by binding the peptides.

12. Line 456: The process should be more detailed due to the amount of peptides synthesized. Is it automated synthesis? (reference of the device used). If it was manual synthesis, which resin was used? controls for each coupling? Which column was used for purification?

This information has now been included in the methodology.

13. Line 463: cleaved with TFA?

This information has now been included in the methodology.

14. Line 490: ¿what kind of broth?

This information has now been included in the methodology.

15. Lines 493-495: OD600 of 0.8 can be equivalent to a greater number of bacteria than that reported since only an OD600 of 0.1 is equivalent to approximately 1×10^8 CFU / ml [1].

We calibrated OD600 and confirmed concentrations of CFU/ml to ensure that final cell count in each well was 5×10^5 CFU/ml before incubation.

16. Line 503: This culture conditions were used for *C. albicans*?

Culture conditions only differed between *C. albicans* and the two bacteria in the broth and agar used. Temperature and time and other time conditions were kept the same across all organisms.

17. Line 506: The IC50 would not be a good indicator to start with the MBC tests due to the variability of this indicator. After 20 hours of incubation, it is very likely that 50% of live bacteria will grow exponentially. Furthermore, the number of CFU and IC50 can vary significantly in an interval of 2 hours (18-20h) since the bacteria duplicate every 30 minutes, providing very imprecise results. Bacterial growth time should be strictly controlled.

We thank the reviewer for spotting this point: we have not explained our methods sufficiently clearly. To clarify we did not use IC50 as an indicator to start the MBC tests. IC50 was calculated independently at the end of the experiment. MBC tests were performed on every well in the plate using an automated method, rather than simply carrying out MBC tests on wells near the IC50 point. We have now amended the methods section to make this clearer.

Report the Minimum Inhibitory Concentration (MIC) is suggested since the MIC is widely accepted and is considered the “gold standard” for determining the susceptibility of organism to antimicrobials and is therefore used to judge the performance of all other methods of susceptibility testing [2]. In addition, the MIC is listed as the starting point for larger pre-clinical evaluations of novel antimicrobial agents [3]. Follow the CLSI guide would be appropriate [4].

We thank the reviewer for raising the issue of the IC50 vs MBC and MIC. We chose to report IC50 rather than MIC as given the low activity of many of the shorter peptides in the library, we wanted to use an approach able to provide a quantifiable value, allowing slightly active peptide to be distinguished from non-active peptides. Had we only reported MIC, as would be appropriate for a study intending to identify the most active leads for subsequent drug development, we would've been unable to distinguish activity between the less active sequences. Our aim was to characterize activity across this peptide space rather than identify those peptides with the strongest activity alone.

18. Line 521: How many erythrocytes would be undergoing the treatment?

Typical erythrocyte concentrations were around 5×10^7 cells /ml. We have now added this information to the text.

Line 531: ¿Positive control used?

Information on positive and negative controls have now been included in the manuscript.

Equation employed for calculate EC50?

As with IC50, EC50 was calculated using a sigmoidal function. This information has now been included in the manuscript.

Line 547: Mention the dual angle used.

We have added this to the methods section.

19. Line 571: What was the temperature used for the preparation?

Information on temperature has been included in the manuscript.

20. Line 577: Small unilamellar vesicles SUVs are characterized by being very unstable and tend to aggregate to form MLVs over time. Were size characterizations performed before being used in biophysical techniques?

DLS was performed before use to ensure that SUVs had not aggregated. This information has been added to the manuscript.

21. Line 616: Why was this bacterium chosen?

We once again chose to use *P aeruginosa* in these experiments as we felt its larger overall size and rod shape might enable more subtle morphological changes induced by the peptides to be visualized, compared to the smaller spherical *S. aureus*. Additionally in preliminary experiments we found that *S. aureus*, when used under the conditions required for the electron microscopy experiments, tended to aggregate which made interpretation of the images more difficult.

22. Line 977: check reference 45.

The reference has been amended.

23. Including in supplementary material a multivariate analysis between charge / amphipathicity and antimicrobial / hemolytic activity would be very interesting.

We agree with the reviewer's helpful suggestion here. We believe these points are addressed by the additional data we have now added in response to R1 comments above, and further by the existing Boruta analysis within the paper.

Reviewer 3.

The current manuscript 'The Lexicon of Antimicrobial Peptides: a Complete Set of Arginine and Tryptophan Sequences' by Clark et al tested antimicrobial and hemolytic functions a complete set of short peptides, composed of arginine and/or tryptophan residues. Several similar works have been reported as cited in the manuscript. One of very important paper (Zhang Liu, Anna Brady, Anne Young, Brian Rasimick, Kang Chen, Chunhui Zhou, Neville R Kallenbach, Antimicrob Agents Chemother. 2007;51(2):597-603. Length effects in antimicrobial peptides of the (RW)_n series) also reported similar idea to design and screen optimal antimicrobial peptides, but the current manuscript does not cite the paper. In the paper of 2007, they have found the shortest antimicrobial peptides with the length of two residues. Given some previous reports, I do not think that the current work has provided novel findings and new advances.

We are grateful to the reviewer for drawing our attention to this interesting paper which we now cite in our article.

We do however respectfully disagree with R3's comments on the relevance of this paper.

The reference cited contains data for only *three* peptides found within our library of 256 peptides. Measuring activities for these three peptides (RW, RWRW, and RWRWRW) is in no way comparable to our study of all possible permutations of R and W, as subtle effects of different arrangements of these amino acids within the peptides could not be examined.

Interestingly this paper did report activity in the peptide containing only 2 residues (RW) – however the IC₅₀ concentration was extremely high (2.1 to 4.3mM), and it is unclear whether at such high concentrations the vehicle control itself would have antibacterial activity. We have however now cited the study, as it did previously measure antimicrobial activity at this extremely high concentration (far higher than the range we and others would typically test). This suggests shorter peptides may have activity if used in the mM range, although care would be needed to distinguish such apparent activity from that of the salt compounds use in peptide synthesis.

Reviewers' comments:

Reviewer #1 (Remarks to the Author):

In their revised paper, the authors have responded satisfactorily to the points raised in my initial review. Therefore, assuming the other reviewers concur, this paper is recommended for publication in Communications Biology in its present form.

Reviewer #2 (Remarks to the Author):

Thank you very much for considering most of the comments. The manuscript has improved remarkably. However, some specific details should be addressed.

Reviewer #3 (Remarks to the Author):

The same concerns as mentioned in my previous comments 'Length effects in antimicrobial peptides of the (RW)_n series) reported similar idea to design and screen optimal antimicrobial peptides in several previous reports'. The current work has not provided novel idea to design and screen optimal AMPs, they only synthesized more peptides (256), but it is easy to predict that most of the peptides are meaningless. For example, the peptides only contain R or W, or very short peptides, which have been demonstrated to be ineffective many times. Thus, among the 256 peptides, most of them are used for only adding the number.

In addition, they did not find any promising candidates. The MBC for the most effective peptide in this work is 65 μ M (>70 μ g/ml), which is much worse than reported candidates with MBCs <1 μ g/ml. It demonstrated that the current method is ineffective.

The Reviewer has helpfully drawn attention to two issues that were not sufficiently clear in the present draft of the manuscript, and which we have now addressed (in reverse order) below. We have also corrected several small typos at various points in the manuscript, and made the abstract slightly clearer.

1. Overall activity of peptides in our Library.

Reviewer 3 Comment: The MBC for the most effective peptide in this work is 65 μ M (>70 μ g/ml), which is much worse than reported candidates with MBCs <1 μ g/ml.

This statement highlights an issue with the way in which we stated the activity of our peptide set within the text of the results section. We quoted the interquartile range around the median value, not the wider range of values (which were reported in the Figures). As such, R3 came away with the false impression that the lowest MBC was only 65 μ M, and that most peptides were inactive.

- In fact the large majority of our peptides showed activity, with 80.3% of sequences showing an IC₅₀ concentration against at least one organism.
- The lowest MBC concentration we found was 9.6 μ M, with many peptides showing MBC values well below the 65 μ M, quoted by R3.

We have now re-written this section of the results, so that the text more accurately reflects the activities apparent in the Figures.

Proposed Change:

Results:

Overall, antimicrobial activity was highest for *S. aureus* > *C. albicans* > *P. aeruginosa*, with most peptides in the heptamer group (which comprised approximately half the library) showing activity. Minimum heptamer IC₅₀ concentrations were 1.9 μ M, 10.3 μ M and 15 μ M for *S. aureus*, *C. albicans* and *P. aeruginosa*, with 92.2%, 81.3% and 60.2% of heptamers showing measurable IC₅₀ concentrations against these three organisms (Fig 1a). MBC values followed a similar pattern with the minimum MBC values for heptamers being 9.6 μ M, 18.7 μ M and 42 μ M for *S. aureus*, *C. albicans* and *P. aeruginosa* respectively, with 60.2%, 60.2% and 10.2% of heptamers showing measurable MBCs against these three organisms (Extended Data Fig. 1a). We found a strong correlation between IC₅₀ and MBC for each of the three organisms for the entire peptide set (*S. aureus*: $\rho = 0.73$, $P < 2.2 \times 10^{-16}$, $S_{129} = 101042$; *C. albicans*: $\rho = 0.93$, $P < 2.2 \times 10^{-16}$, $S_{114} = 17027$; *P. aeruginosa*: $\rho = 0.78$, $P = 2.375 \times 10^{-5}$, $S_{19} = 384.42$).

Discussion:

Although a measurable MBC is a key attribute of an AMP, as we focused on investigating the development of antimicrobial activity in this peptide set, we also considered IC₅₀ values to be important as an indicator. Indeed we found that IC₅₀ concentrations were likely to be a good indicator of activity overall, since the IC₅₀ and MBC against each organism was highly correlated for peptides in the set which exhibited activity in both metrics.

2. The Overall Aim of the Study.

Reviewer 3 Comment: The same concerns as mentioned in my previous comments 'Length effects in antimicrobial peptides of the (RW)_n series' reported similar idea to design and screen optimal antimicrobial peptides in several previous reports'.

The current work has not provided novel idea to design and screen optimal AMPs, they only synthesized more peptides (256), but it is easy to predict that most of the peptides are meaningless. For example, the peptides only contain R or W, or very short peptides, which have been demonstrated to be ineffective many times. Thus, among the 256 peptides, most of them are used for only adding the number.

In addition, they did not find any promising candidates.

It demonstrated that the current method is ineffective.

It seems that R3 had formed the impression that the aim of our study was to test a new method to directly enable and demonstrate the identification of 'promising candidates'. Indeed the few previous studies to look at peptides, focused on very small numbers of peptides, as part of a wider optimisation process to directly identify more active sequences. These were not systematic, and covered only a minuscule proportion of the possible peptide space.

In fact our aim was completely different. Instead we wished to uncover how activity develops differentially across this peptide space, as these two individually inactive amino acids are combined in every possible permutation up to and including the heptamer sequences.

We have published a number of studies on larger (12 – 18 amino acids) AMPs, with greater activity overall (Dobson et al., 2006; Kelly et al., 2007, 2010; Forbes et al., 2013), and like others have found that longer peptides are more active. However their larger size means that it is almost impossible to generate a meaningful proportion of possible variants and truly begin to understand how and why activity develops, and identify those primary structural features that confer activity.

Our study featured two principles that together enabled us to achieve our objectives:

- **Complete Set.** It is only possible to understand the primary structural basis for activity by studying a complete set of peptides – as such none of our findings need to be extrapolated within the peptide space we are working as all activity has been surveyed.
- **Only studying R and W.** Of the naturally occurring amino acids, R and W are the most cationic and hydrophobic respectively, and therefore the amino acids most central to the development of antimicrobial properties. It would have been a mistake to try to add additional residues, as (i) we would not then be able to study a complete set of peptides, and (ii) overall the activities of the most active peptides would have been lower.

We expected and welcomed uncovering some inactive sequences, and identifying those peptides with sequence features which proved not to be favourable to generate activity, thus enabling us to establish fundamental rules for activity by contrasting these with sequences which were moderately or highly active.

An unsuccessful outcome for our study would have been to find that all 256 peptides were equally active (or inactive), but we did not find this. Instead we found great diversity in activity across the peptide set, which was the perfect outcome, enabling us to establish some general rules:

- Increased activity is found in sequences with around 40% R.
- W doublets and triplets increase activity relative to W alone.
- The peptides activity as a whole involves membrane binding to negatively charged microbial membrane and furthermore activity due to peptides developing an aggregated form.
- Peptides > 10 residues are likely to reduce activity per residue of peptide.

Overall our study of the complete set of peptides built from the two most relevant amino acids for developing antimicrobial activity provides a framework to develop insights into fundamental broadly applicable rules applicable to all AMPs. This provides opportunities for further studies to build on this study considering other residue combinations, and to explore how rules developed using these peptides with emerging activity translate to predictions in larger peptides.

Proposed Change:

To further strengthen the manuscript, and clarify that our aim was to use an innovative approach to uncover fundamental understanding rather than identify longer active clinical leads we have included the following additions to the opening paragraph of the results and the discussion:

Results:

We considered all possible sequences of W and R, up to 7 residues long. **These two amino acids are closely associated with AMP activity, however, the shortest 'peptides' assayed were the individual amino acids, which we expected to be inactive. In this way we uncovered the development of antimicrobial activity in fine detail.**

Discussion:

Although previous studies have provided anecdotal examples of the impact of sequence changes on the activity of several peptides²³, we examine these effects using a *complete* set of all peptides possible within a theoretical universe, and therefore we can more reliably make broad conclusions at least for the peptide class studied. It should be noted that our aim was to provide some fundamental insights into the rules governing the development of activity in this class of peptides, not to directly identify potential clinical lead compounds, for which further considerations apply²⁴.

24 Browne, K. *et al.* A New Era of Antibiotics: The Clinical Potential of Antimicrobial Peptides. *Int. J. Mol. Sci.* **21**, 7047 (2020).

References:

Dobson, C.B. *et al.*, 2006. The receptor-binding region of human apolipoprotein E has direct anti-infective activity. *The Journal of infectious diseases*, 193(3), pp.442–450.

Forbes, S. *et al.*, 2013. Comparative surface antimicrobial properties of synthetic biocides and novel human apolipoprotein E derived antimicrobial peptides. *Biomaterials*, 34(22), pp.5453–5464.

Kelly, B.A. *et al.*, 2007. Apolipoprotein E-derived antimicrobial peptide analogues with altered membrane affinity and increased potency and breadth of activity. *The FEBS journal*, 274(17), pp.4511–4525.

Kelly, B.A. *et al.*, 2010. Anti-infective activity of apolipoprotein domain derived peptides in vitro: identification of novel antimicrobial peptides related to apolipoprotein B with anti-HIV activity. *BMC Immunology*, 11(1), p.13.

REVIEWERS' COMMENTS:

Reviewer #2 (Remarks to the Author):

Comments have been sent to the editor

Reviewer #3 (Remarks to the Author):

Generally, I do not think the the idea in the MS is not over previous works such as Length effects in antimicrobial peptides of the (RW)_n Series. Antimicrob. Agents Chemother. 2007, 51, 597–603; Antifungal activity of (KW)_n or (RW)_n Peptide against *Fusarium solani* and *Fusarium oxysporum*. Int. J. Mol. Sci. 2012, 13, 15042–15053; The pharmacophore of short cationic antibacterial peptides. J Med Chem 2003, 46:1567–1570. Eespecially, in the report of JMC 2003, a set of peptides containg only W and R have been intensively studied.

WRWRWR-NH₂

RWRWRW-NH₂

RRRWWW-NH₂

RWWWRR-NH₂

WRRRRW-NH₂

WRWRW-NH₂

RWRWR-NH₂

WRYRW-NH₂

WRWRY-NH₂

WRWR-NH₂

WRRW-NH₂

RWWR-NH₂

WRW-NH₂

RWR-NH₂

RWWR-OMe

RWRW-OBzl

RWRw-OBzl Capital letters denote L-amino acids, and lower case letters denote D-amino acids

WRWR-OMe

WWR-OMe

WRW-OBzl

wrw-OBzl

wRW-OBzl

WR-OMe

WR-OBzl

RW-OBzl

Rw-OBzl

rW-OBzl

rw-OBzl

KW-OBzl

Kw-OBzl

kW-OBzl

RF-OBzl

FR-OBzl

KF-OBzl

As to the rules deduced by the authors:

1, Increased activity is found in sequences with around 40% R.

It is not new because it is well known that 2-3 R residues in the peptides containing 4-7 amino acid residues keep necessary net positive charges and hydrophobicity endowed by several W residues. More R will results less W or Less R results more W, which possibly destroys the antimicrobial ability.

2, W doublets and triplets increase activity relative to W alone.

It is not new because it is well known that it is necessary for any antimicrobial peptides to contain enough hydrophobicity. Doublets and triplets increase activity relative to W alone because they increase the hydrophobicity.

3, The peptides activity as a whole involves membrane binding to negatively charged microbial membrane and furthermore activity due to peptides developing an aggregated form. It is not new because they have been demonstrated for many times.